# Evaluating the E3SMv2-MPAS ocean-sea ice coupled unstructured model in the Arctic: Atlantification processes and systematic biases

Xinyuan Lv<sup>1</sup>, Huizan Wang<sup>1</sup>, Yu Cao<sup>1</sup>, Kaijun Ren<sup>1</sup>, Yangjun Wang<sup>2</sup> and Hao Ding<sup>1</sup>

<sup>1</sup>College of Meteorology and Oceanology, National University of Defense Technology, Changsha, 410073, China <sup>2</sup>College of Advanced Interdisciplinary Studies, National University of Defense Technology, Nanjing, 211101, China *Correspondence to*: Huizan Wang (wanghuizan@126.com) and Kaijun Ren (renkaijun@nudt.edu.cn)

Abstract. Advancing high-resolution Arctic ocean-sea ice modeling is critical for understanding polar amplification and improving climate projections but faces challenges from computational limits and cross-scale interactions. The simulation capabilities of the ocean-sea ice coupled model (E3SMv2-MPAS) from the Energy Exascale Earth System Model (E3SM) 2.1 for the Arctic ocean-sea ice system are systematically evaluated using multi-source observational data. The model employs a latitudinally varying mesh, with resolution increasing from 60 km in the Southern Hemisphere to 10 km in the Arctic. This design balances computational efficiency with the accurate integration of low-latitude oceanic influences, while the unstructured mesh also enhances the geometric representation of Arctic straits. Together, these features form a simulation framework capable of resolving processes from seasonal to decadal timescales. Numerical results demonstrate E3SMv2-MPAS's superior Arctic simulation performance: (1) Accurate reproduction of spatial heterogeneity in sea ice concentration, thickness, and sea surface temperature, including their 1995-2020 trend patterns; (2) Faithful reproduction of both the freshwater content and transports through key Arctic gateways; (3) Successful reconstruction of three-dimensional thermohaline structures within the Atlantic Water layer, capturing Atlantic Water's decadal warming trends and accelerated Atlantification processes – specifically mid-layer shoaling, heat content amplification, and reduced heat transfer lag times in the Eurasian Basin. Persistent systematic biases are identified: 0.5-1 m sea ice thickness overestimation in the Canadian Basin; Coordinated sea surface temperature/salinity underestimation and sea ice concentration overestimation in the Greenland and Barents Seas; Atlantic Water core temperature overestimation; Regional asymmetries in decadal thermohaline field evolution.

#### 1 Introduction

The Arctic region has emerged as one of the most rapidly transforming area of the Earth system under contemporary climate change (Calvin et al., 2023). However, persistent gaps in oceanic observational networks, particularly the lack of systematic full-depth and pan-strait measurements across key Arctic gateways, have significantly constrained our understanding of Arctic oceanic transport dynamics. To address these observational limitations, numerical modeling has become an indispensable tool (Wang et al., 2023). Of particular scientific significance is the thermohaline transport through Fram Strait

- the principal conduit for Atlantic Water (AW) intrusion into the Arctic basins (Fu et al., 2023; Karami et al., 2021; Long et al., 2024). Recent studies highlight the necessity to quantify both the spatiotemporal evolution of AW-derived heat distribution across Arctic marginal seas and the relative contributions of different vertical heat flux mechanisms (Carmack et al., 2015; Polyakov et al., 2020b). State-of-the-art global climate models (GCMs) provide critical insights into the evolving climate system under sustained global warming scenarios, enabling the investigation of multi-sphere interactions and their associated feedback mechanisms (Dörr et al., 2021; Hinrichs et al., 2021; Rieke et al., 2023; Shu et al., 2022).

While climate models remain indispensable tools for deciphering Earth system dynamics (Landrum and Holland, 2020). their representation of Arctic processes exhibits persistent uncertainties that challenge predictive capabilities (Pan et al., 2023). Systematic biases plague the simulation of critical Arctic phenomena, including amplified warming rates, sea ice retreat patterns, and AW layer evolution (Heuzé et al., 2023; Khosravi et al., 2022; Muilwijk et al., 2023; Shu et al., 2019). These limitations persist across successive model generations, as evidenced by Coupled Model Intercomparison Project Phase 5 (CMIP5) and Phase 6 (CMIP6) revealing substantial errors in Arctic three-dimensional thermohaline structure reproduction (Khosravi et al., 2022; Shu et al., 2019). There are mainly four common biases of contemporary models in the Arctic include: (1) Overestimated AW layer thickness and depth. This systematic vertical structure misrepresentation persists across model generations, from early Arctic Ocean Model Intercomparison Project (AOMIP) simulations (Holloway et al., 2007) through the Coordinated Ocean-ice Reference Experiments, phase II (CORE-II; Ilicak et al. (2016)), to the most widely used CMIP5/CMIP6 ensembles (Heuzé et al., 2023; Khosravi et al., 2022; Shu et al., 2019). Among 41 CMIP5 models evaluated by Shu et al. (2019), 22% failed basic AW identification criteria, while the remaining 32-model mean overestimated AW layer vertical extent compared to observational benchmarks. CMIP6 shows limited improvement, with multi-model mean AW upper boundaries erroneously positioned at ~400 m depth in the Nansen Basin – deeper than observed values – and excessive thickness extending to the seafloor in some regions (Khosravi et al., 2022). (2) Cold bias in AW core temperatures. The Alfred Wegener Institute coupled climate model (AWI-CM1) exhibits thermal underestimation at 200-600m depths in Eurasian Basin simulations (Hinrichs et al., 2021), consistent with CMIP6's 0.4°C cold bias relative to hydrographic climatologies (Heuzé et al., 2023). (3) Failure to capture AW warming trends. CMIP5 models collectively underestimate observed decadal temperature variability, with no model replicating the post-2000 acceleration in AW

increasingly akin to the warmer and saltier AW). While models project gradual boreal water encroachment in the Barents Sea by 2100 (Wassmann et al., 2015), observational analyses suggest this regime shift is likely to occur at a faster pace (Lind et al., 2018). Discrepancies extend to sea ice thermodynamics, where Seasonal Forecast System 5 (SEAS5) simulations yield only 10–20 cm winter ice production decline (Polyakov et al., 2022), versus 78–93 cm observed losses (Polyakov et al.,

warming (Shu et al., 2019). (4) Underestimated "Atlantification" (referring to the Arctic Ocean water properties becoming

2020b).

There are numerous and complex reasons that lead to the common deviations in models when simulating the AW. These challenges can be categorized into four primary domains: (1) Insufficient horizontal resolution (>50 km in most CMIP6 models) fails to resolve critical boundary currents and mesoscale eddies (Hinrichs et al., 2021); (2) Unrealistic Atlantic-

Arctic exchange through Fram Strait (Hinrichs et al., 2021); (3) Parameterization deficiencies, including the incorrect representation of horizontal advection and vertical mixing (Lind et al., 2018); (4) Imperfect knowledge of ocean-sea ice-atmosphere triadic feedbacks, especially during winter convection events, hampers accurate simulation of AW ventilation processes (Heuzé et al., 2023). To advance model fidelity and reduce uncertainty sources, comprehensive investigations into systematic model biases are imperative (Hinrichs et al., 2021; Pan et al., 2023).

Current numerical simulations for polar regions are primarily based on structured grid models. However, the inherent limitations of structured grids, particularly the singularity at the North Pole and meridional convergence artifacts, fundamentally constrain their capacity to represent Arctic-specific physical processes (Liu et al., 2016). These geometric constraints not only distort parameterization schemes but also introduce systematic biases in both regional and decadal-scale simulations. While global high-resolution configurations could theoretically mitigate such issues, their prohibitive computational costs render them impractical for climate-scale applications (Golaz et al., 2019). This technological impasse has driven the development of two complementary approaches: (1) Nested grid systems: Though offering advantages in spatial discretization flexibility and geometric simplification, their implementation introduces nontrivial challenges in mass conservation, interface coupling fidelity, and numerical noise suppression (Hoch et al., 2020). (2) Unstructured mesh: By enabling localized resolution enhancement in dynamically critical zones while maintaining coarse resolutions elsewhere, these meshes eliminate the need for explicit nesting procedures (Scholz et al., 2019). Their continuous spatial adaptability allows direct resolution of sub-mesoscale processes without compromising computational efficiency (Wang et al., 2018).

The application of variable-resolution models with a global unstructured mesh offers distinct advantages for Arctic Ocean studies. By employing high-resolution meshes over the Arctic region, these configurations enable accurate simulation of energy exchange processes across narrow critical channels (e.g., Fram Strait, Bering Strait, Barents Sea Opening and Davis Strait). Coarser resolutions in other domains maintain computational efficiency while preserving connectivity between the Arctic and extratropical regions (Wang et al., 2018). Among global implementations, two widely adopted models are the Finite-Volume Coastal Ocean Model (FVCOM; Chen et al. (2016)) and the Finite-Element Sea ice-Ocean circulation Model (FESOM; Danilov et al. (2017)). In Arctic studies, FVCOM predominantly operates as a regional model, as evidenced by its frequent implementation in localized domains (e.g., Zhang et al. (2016)). This regional focus aligns with FVCOM's original design paradigm prioritizing coastal and shelf-sea dynamics through its finite-volume discretization scheme. In contrast, FESOM has been predominantly implemented as a global model in Arctic studies, where its implementation has demonstrated unprecedented skill in simulating Arctic intermediate water dynamics (Danilov et al., 2017; Wang et al., 2018; Wekerle et al., 2013). Notably, Wang et al. (2018) established that FESOM (a relatively low resolution, ~24 km in the Arctic) outperforms a set of the state-of-the-art structured-grid models evaluated by Ilicak et al. (2016), particularly in correcting systematic AW core biases.

As a more recent modeling framework relative to FESOM and FVCOM, the Model for Prediction Across Scales (MPAS) remains in the nascent phase of Arctic performance evaluation (Huo et al., 2024; Ringler et al., 2013), particularly regarding its capacity to simulate intermediate water masses and Atlantification processes. The Energy Exascale Earth System Model

(E3SM), evolved from the Community Earth System Model (CESM), incorporates MPAS-Ocean and MPAS-Seaice as its ocean and sea ice components. Initial assessments using E3SMv1's ocean-sea ice coupled configuration (60to10 km variable resolution) demonstrate promising skill in reproducing pan-Arctic freshwater budgets, gateway current exchanges, and vertical hydrographic profiles (Veneziani et al., 2022). Persistent errors in sea ice thickness (SIT) distribution and upper 100 m stratification emerge across resolutions, suggesting common structural model deficiencies rather than discretization artifacts. However, their diagnostic lack the rigorous validation metrics employed by Wang et al. (2018) for FESOM's AW representation. Existing assessments predominantly rely on pan-Arctic-basin-averaged diagnostics, obscuring critical vertical and regional heterogeneities in intermediate AW layer dynamics (Veneziani et al., 2022).

This study presents a tripartite evaluation framework for the coupled system of MPAS-Ocean and MPAS-Seaice in E3SM version 2 (E3SMv2-MPAS), which compares it with the observational datasets and reanalysis products to systematically assess MPAS's capacity in simulating key Arctic processes. In addition, we conduct a comprehensive assessment of Arctic sea ice dynamics, surface layer hydrographic properties, three-dimensional thermohaline profile evolution (particular emphasis on the AW layer), as well as freshwater content and key gateway transports. The assessment highlights the model's strengths, identifies its limitations, and discusses potential sources of uncertainty. Innovatively, this work implements a multi-layer connectivity analysis examining cross-layer interactions between surface (10 m) and intermediate (400 m) depths.

The subsequent sections are structured as follows: Section 2 provides comprehensive documentation of the E3SMv2-MPAS configuration and validation datasets. Section 3 and Section 4 conduct rigorous multi-faceted analyses of Arctic-specific simulations, employing both domain-wide diagnostics and sub-regional decomposition approaches. Section 5 discusses the potential advantages of higher resolution and unstructured meshes, summarizes simulated biases and their possible sources, and identifies limitations in our model design and configuration. Finally, Section 6 synthesizes the key findings and outlines broader implications.

#### 2 Model configurations and data

## 2.1 Model configuration






Veneziani et al. (2022) demonstrated that refining mesh resolution from 10 km to 6 km triples computational costs without yielding significant improvements in simulation fidelity. Their findings suggest that resolving the local Rossby radius of deformation across most Arctic regions necessitates resolutions ≤3 km − a requirement currently constrained by prohibitive computational demands. The model configuration in this paper is described as follows. To address the trade-off between high-resolution requirements and computational constraints, our study employs a variable-resolution unstructured mesh featuring a meridional transition from 60 km resolution in the Southern Hemisphere to 10 km in the Arctic domain (hereafter 60to10 km, same as Veneziani et al. (2022); Fig. 1a). This adaptive meshing approach optimizes computational efficiency while resolving critical processes: (1) Antarctic coastal regions (80° S−90° S) maintain a 25 km resolution to capture fine-

scale dynamics; (2) the North Atlantic sector is strategically refined, transitioning from 20 km to 10 km resolution earlier than the Pacific to guarantee at least 15 km resolution in the Gulf Stream extension region (~40° N; Veneziani et al. (2022)); (3) the North Pacific sector maintains computational efficiency while achieving approximately 10 km resolution in its subpolar region adjacent to the Arctic Ocean (north of 50° N).

Figure 1. (a) Geographical distribution of grid cell size (km) of the E3SMv2-MPAS framework. (b) bathymetry from the ETOPO 2022 and key basins/straits north of 60° N. EEB and WEB refer to the eastern and the western Eurasian Basin respectively. The black dashed transect along 70° E and 145° W (crossing the North Pole) denotes the location of the transect shown in Fig. 14.




Numerical stability was achieved through a 5-minute baroclinic time step for ocean dynamics. For sea ice, we employed a 15-minute dynamic time step and a 30-minute thermodynamic time step (a 2:1 ratio). MPAS-Ocean adopts finite volume discretization of primitive governing equations within a staggered C-grid framework, incorporating hydrostatic, incompressible, and Boussinesq approximation assumptions (with a z-star vertical grid) (Golaz et al., 2019). Vertical mixing processes were parameterized using the K-profile scheme (KPP; Large et al. (1994)). For mesoscale eddy representation, similarly to what was done in Veneziani et al. (2022), we implemented a spatially varying Gent-McWilliams (GM) parameterization, incorporating both bolus advection and Redi isopycnal diffusion components (Gent and Mcwilliams, 1990). The eddy diffusivity coefficient (κ) was given a latitudinal dependence: 300 m<sup>2</sup> s<sup>-1</sup> in high-resolution Arctic regions (<20 km grid spacing) to maintain moderate mixing intensity, transitioning linearly to 1800 m<sup>2</sup> s<sup>-1</sup> in low-resolution zones (>30 km grid spacing) to compensate for unresolved eddy fluxes (Fig. 2). MPAS-Seaice builds upon the core numerical and physical framework of the Los Alamos Sea Ice Model (CICE). The dynamics are governed by the elastic-viscous-plastic (EVP) rheology, with the internal ice stress divergence operator adapted for MPAS's unstructured polygonal mesh (Turner et al., 2022). Sea ice and tracer transport are handled by an incremental remapping scheme (Lipscomb and Ringler, 2005), adapted for polygonal cells. The thermodynamics and vertical column physics remain consistent with CICE (Turner et al., 2022). The configuration includes the "mushy layer" thermodynamics for vertical heat transfer, the delta-Eddington shortwave radiation scheme, a level-ice melt pond parameterization, ice thickness distribution mechanics, and transport in thickness space (Petersen et al., 2019). The specific configurations of MPAS-Ocean and MPAS-Seaice within the E3SMv2, including their coupling mechanisms, have been comprehensively documented in Turner et al. (2022), Golaz et al. (2022) and Huo et al. (2024).

In addition to the ocean and sea ice components, the atmospheric and river modules in E3SMv2-MPAS (see Fig. 2 for specific variables used) were forced by the JRA55-do (v1.5; Tsujino et al. (2018)) from the Japan Meteorological Agency (JMA). This dataset has high spatiotemporal resolution (3-hourly temporal and 0.5625° spatial resolution) and spans the period from 1958 to 2020. Sea surface salinity (SSS) was relaxed toward Polar science center Hydrographic Climatology (PHC) 3.0 climatology (Steele et al., 2001) with an annual restoring timescale.





Figure 2. Configuration details for E3SMv2-MPAS: forcing/initial conditions, runtime settings, and output fields.

Given the prohibitive computational cost of a continuous high-resolution simulation from 1958 to 2020, we adopted a strategic two-period integration scheme to prioritize computational resources for our core analysis period (1995–2020). The model's climatological fidelity during this satellite era is verified using multi-source observational data, ensuring a reliable assessment of both sea ice and ocean variability.

The MPAS-Ocean component was initialized from a pre-processed state (ocean.ARRM60to10.180715.nc). This state was derived from a prior short-term (5-day) adjustment run of the standalone ocean model, which itself started from a state of rest with three-dimensional temperature and salinity fields prescribed from the PHC. Consequently, this initial condition provided a dynamically adjusted and physically consistent starting point for our coupled simulation, mitigating the initial shock that would otherwise occur from a purely cold start. In contrast, the MPAS-Seaice component was initialized from an idealized, uniform ice cover. A 1-meter thick ice layer with 100% concentration was prescribed on all ocean grid points between 60° S and 70° N, with zero initial snow depth and stationary ice velocity. This simple state allowed the sea ice cover to evolve self-consistently in response to the model's atmospheric forcing and ocean coupling from the beginning of the simulation. Following this spin-up phase, the full interannual JRA55 forcing was applied from 1958 to 1981.

To begin the simulation for our main analysis period (1995–2020), we used the model state from December 1981 as the initial conditions for January 1995. This 13-year gap (1982–1994) was a strategic choice to conserve computational resources while ensuring physical consistency in the key variables of interest. This computational strategy was motivated by the fact that, under forcings such as CORE-II or JRA55 and when initialized with PHC hydrography, upper-ocean and surface variables are known to reach quasi-equilibrium within a few decades, as demonstrated in several previous studies. For instance, Wang et al. (2018) reported that temperature and salinity in the upper 1000 m reached near-equilibrium within 20–30 years. Wekerle et al. (2013) began their analysis of surface variables and freshwater content in the 0–500 m layer after a 10-year initialization in a 1958–2007 simulation using FESOM under CORE-II forcing. Likewise, in the analysis of multiple high-resolution Ocean Model Intercomparison Project Phase 2 (OMIP2) models simulating the full 1958–2020 period under JRA55 forcing, Wang, Shu, Bozec, et al. (2024) focused their evaluation on the period 1971–2000 – commencing approximately 13 years after the model initialization. In our simulation, the 24-year spin-up from 1958 to 1981 is largely sufficient for the adjustment of surface fields (e.g., sea ice, surface temperature, and salinity) and AW layer (above 1000 m), which are the focus of this study. Although the deep ocean remains far from equilibrium, the targeted variables had largely stabilized by 1981.






From a physical perspective, the potential impact of this initialization approach for the 1995–2020 simulation is expected to be short-lived. The upper ocean and sea ice (the primary focus of this study), adjust much more rapidly than the deep ocean, and their evolution is predominantly governed by contemporaneous atmospheric forcing rather than by the initial conditions. Therefore, the disequilibrium introduced by the initial condition from 1981 would be rapidly overwritten and adjusted by the realistic, synchronous atmospheric forcing applied from 1995 onward.

Therefore, initializing the 1995 run from the 1981 output allows a computationally efficient hot start and ensures that the model is in an appropriate state for evaluating the 1995–2020 period.

The model output initialized from the 1981 state also demonstrates physically consistent behavior during the 1995–2020 period, further supporting the validity of this approach. The temporal evolution of key diagnostic variables – including sea surface temperature (Fig. 8d) and sea ice-related variables (Fig. 7) – shows that the simulation quickly aligns with the observed/reanalysis trajectory after 1995, with no persistent systematic bias. Spatial distributions of these variables are also in good agreement with evaluation datasets (Figs. 3–5, 8a–c), and the long-term trends from 1995 to 2020 closely match those in the references (Fig. 7). These results, which will be discussed in detail in the following sections, indicate that the initialization from 1981 did not adversely affect the simulation of central climate features during the study period.

Accordingly, our primary evaluation focuses on the performance of E3SMv2-MPAS during the period 1995–2020. In addition, a comparative assessment of the 1960–1980 period is also included to briefly examine the decadal variability of key ocean and sea ice variables and to verify the model's capability under distinctly different climatic backgrounds.

## 2.2 Evaluation datasets

## 2.2.1 Sea ice concentration, extent, thickness, and volume

- To comprehensively evaluate sea ice concentration (SIC) performance, both the observations and reanalysis data were adopted for validation. SIC datasets used here include: (1) Passive microwave remote sensing data: Sourced from the NOAA/NSIDC Climate Data Record (Version 4; Meier et al. (2021)) with a spatial resolution of 25 km × 25 km; (2) HadISST1 data: Provided by the UK Met Office Hadley Centre (Rayner et al., 2003) at 1° × 1° resolution; (3) ERA5 reanalysis: Generated by the European Centre for Medium-Range Weather Forecasts (ECMWF; Hersbach et al. (2020)) at 0.25° × 0.25° resolution.
- For SIT validation, we utilize four key datasets: (1) Pan-Arctic Ice-Ocean Modeling and Assimilation System (PIOMAS; Zhang and Rothrock (2003)): This reanalysis product, extensively validated against satellite and in situ observations, provides reliable Arctic SIT spatial distributions and long-term trends (Laxon et al., 2013; Schweiger et al., 2011; Stroeve et al., 2014). (2) PIOMAS-20C reanalysis (Schweiger et al., 2019): Driven by ECMWF's atmospheric reanalysis of the 20th century (ERA-20C) and calibrated with historical in situ/aircraft measurements, this dataset enables analysis of pre-satellite-era SIT variability (1960–1980). (3) CS2SMOS gridded product: developed by the Alfred Wegener Institute (AWI) and the University of Hamburg (Ricker et al., 2017), it combines CryoSat-2 and SMOS satellite observations using an Optimal Interpolation method. The data cover the period from October to April each year, when the sea ice is more stable, thereby minimizing signal interference from summer melt ponds and enhancing the reliability and accuracy of the dataset.
- For sea ice extent (SIE), the evaluation dataset was obtained from the NSIDC (Fetterer, 2017). Sea ice volume (SIV) was assessed using outputs from the PIOMAS and PIOMAS-20C reanalysis.

# 2.2.2 Sea Surface Temperature and Salinity

Sea surface temperature (SST) validation dataset is NOAA's 1/4° Daily Optimum Interpolation Sea Surface Temperature (OISST; Huang et al. (2021)) dataset, which represents a long-term climate data record integrating multi-platform observations from satellites, ships, buoys, and Argo floats. Spatially continuous global SST fields are reconstructed using optimal interpolation to fill data gaps.

For open-water SSS validation (SIC

Wang et al. (2024)'s OMIP2 dataset includes AW core temperature (defined as maximum temperature in water columns over seafloor depths >150 m; 2006–2017) from five high-low resolution model pairs. This dataset is employed to benchmark E3SMv2-MPAS's AW core temperature simulations against multi-model ensembles.

## 3 Arctic physical system states

#### 275 3.1 Sea ice characterization



This study focuses on the Arctic region, systematically evaluating the simulation performance of the E3SMv2-MPAS coupled model for SIC, SIT, SIE and SIV at first. Through comparisons with multi-source observational datasets and reanalysis products, combined with climate-state analysis (1995–2020) and trend diagnostics across two periods (1960–1980 and 1995–2020), model strengths and limitations in polar environmental simulations are identified.

Multi-dataset validation using NSIDC satellite remote sensing (Meier et al., 2021), Hadley in situ assimilation (Rayner et al., 2003), and ERA5 reanalysis (Hersbach et al., 2020) demonstrates that E3SMv2-MPAS effectively captures spatial heterogeneity in Arctic SIC climatology in both winter and summer (Figs. 3 and 4). In winter, consistent spatial bias patterns are observed across datasets, with persistent positive bias center (ΔSIC>0.3) identified along the southwestern Greenland Sea shelf margin and the northern Barents Sea slope (Fig. 3e–g). During summer, comparisons with NSIDC and Hadley reveal predominant positive biases in the northern Barents Sea and widespread negative biases across the central Arctic Ocean (Fig. 4e and f). In contrast, the comparison with ERA5 shows negligible underestimation in the central Arctic basin, while positive biases are observed not only in the northern Barents Sea but also in the Beaufort Sea (Fig. 4g).

Figure 3. During winter (December–February), (a–d) 1995–2020 climatological mean SIC spatial distributions: (a) E3SMv2-MPAS simulations, (b) NSIDC observational product, (c) Hadley Centre HadISST data, (d) ERA5 reanalysis. (e–g) SIC bias fields: (e) E3SMv2-MPAS vs. NSIDC, (f) E3SMv2-MPAS vs. Hadley, (g) E3SMv2-MPAS vs. ERA5.

Figure 4. The same as Fig. 3, but during summer (June-August).



Beyond SIC, SIT serves as a critical parameter governing sea ice dynamics, with its simulation accuracy directly modulating the spatiotemporal heterogeneity of ice volume. We systematically quantify E3SMv2-MPAS's capability in reproducing spatial distribution of SIT during both in winter and summer (Fig. 5). Overall, the model captures the climatological spatial gradient of Arctic SIT, characterized by a gradual thickening from the Barents Sea toward the central Arctic Basin and the northern Canadian Archipelago in both seasons. The model realistically represents the seasonal reduction in SIT over the continental shelf regions along the Arctic margin in summer compared to winter. However, pronounced zonal positive biases (ΔSIT>1.5 m) are present in both seasons, particularly along the eastern and northern shelf of the Greenland Sea, the region north of the Canadian Archipelago, and the southern Canadian Basin and Beaufort Sea (Fig. 5c and f).

Figure 5. During (a-c) winter (December-February) and (d-f) summer (June-August), (a-b and d-e) 1995-2020 climatological mean SIT spatial distributions: (a and d) E3SMv2-MPAS, (b and e) PIOMAS. (c and f) SIT bias field: E3SMv2-MPAS vs. PIOMAS.




Considering PIOMAS's known limitations in overestimating thin ice while underestimating thick ice (Laxon et al., 2013; Schweiger et al., 2011), additional validation using CS2SMOS data (Ricker et al., 2017) is conducted (Fig. S1). Consistent with previous findings, PIOMAS exhibits underestimation in regions with thicker sea ice, such as north of the Canadian Archipelago and east of Greenland (Fig. S1e). Similarly, E3SMv2-MPAS shows pronounced positive biases relative to CS2SMOS in areas including the northern Canadian Archipelago, the southern Canadian Basin, and the Beaufort Sea (Fig. S1d), aligning with the bias pattern identified in comparisons with PIOMAS (Fig. 5c), thereby corroborating the spatial reliability of PIOMAS-indicated biases.

The spatial pattern of maximum positive bias in E3SMv2-MPAS remains consistent across seasons (Fig. 5c and f). This bias is also evident in the model's annual mean SIT distribution relative to PIOMAS (Fig. S2). Previous studies based on CMIP5 models have established a strong correlation between inaccuracies in simulating the Beaufort Gyre and SIT distribution (Stroeve et al., 2014). Since the SSH field serves as a key proxy for evaluating the fidelity of Beaufort Gyre simulations (Wang et al., 2018), we analyze differences in SSH between E3SMv2-MPAS and the ORAS5 reanalysis (Fig. 6a–c). The model overestimates SSH in the Beaufort Sea, suggesting an erroneously enhanced ice convergence. Additionally, the simulated OHC in the 0–100 m layer is underestimated in this region (Fig. 6d–f), which may further contribute to the

positive SIT bias. Thus, the persistent 0.5–1 m positive bias in the Beaufort Sea is hypothesized to originate from an overestimated intensity of the Beaufort Gyre and associated upper-ocean thermal biases in E3SMv2-MPAS, which then may impede the realistic export of sea ice through the north of Canadian Archipelago and east of Greenland.

Figure 6. Climatological mean spatial distributions for the period 1995–2020. (a-c) SSH: (a) E3SMv2-MPAS, (b) ORAS5. (c) Bias in SSH between E3SMv2-MPAS and ORAS5. (d-f) OHC in the upper 100 m: (d) E3SMv2-MPAS, (e) IAP. (f) Bias in OHC between E3SMv2-MPAS and IAP.

To further analyze long-term trends in sea ice-related variables, including interannual and decadal variability, time series of SIC, SIT, SIE, and SIV are examined over the periods 1960–1980 and 1995–2020 (Fig. 7).

Figure 7. Time series and linear trends of sea ice properties from 1960–1980 and 1995–2020. (a) SIC of pan-Arctic (70°–90°N) for E3SMv2-MPAS (gray), NSIDC (blue), Hadley (green), and ERA5 (orange). (b) SIE for E3SMv2-MPAS (black) and NSIDC (red). (c) SIT of pan-Arctic (70°–90° N) for E3SMv2-MPAS (black) and PIOMAS (red). (d) SIV for E3SMv2-MPAS (black) and PIOMAS (red). Dashed lines denote linear trends based on least-squares regression.

E3SMv2-MPAS successfully reproduces SIC seasonal cycles and interannual variability during 1995–2020, maintaining root mean square errors (RMSE) values of 0.040, 0.052, and 0.051 against NSIDC, Hadley, and ERA5 datasets respectively (Fig. 7a). This validates the dynamic framework's effectiveness in capturing sea ice-atmosphere coupling mechanisms. Trend analysis confirms the model's climate response capability. During the rapid decline period (1995–2020), E3SMv2-MPAS accurately captures accelerated SIC reduction trends, showing better agreement with NSIDC observations than Hadley and ERA5 products. For the weak-trend period (1960–1980), the model reproduces quasi-stable sea ice coverage characteristics.

The accelerated SIC decline in the recent period compared to historical decades (1960–1980) highlights the model's ability to replicate trend amplification under intensified forcing, thereby bolstering confidence in its scenario-dependent projections. Similarly, the model effectively captures the interannual and decadal variability of SIE (Fig. 7b; RMSE: 0.96).

Consistent with NSIDC, simulated SIC and SIE exhibit certain seasonal biases. The systematic winter overestimation, attributable to positive SIC biases in the southern Greenland Sea and southward-expanded ice cover in the Barents Sea (Fig. 3e), coinciding with pronounced cold SST biases in these regions (Fig. S3). During summer, E3SMv2-MPAS overestimates the seasonal minimum (Fig. 7a–b), particularly in the Greenland Sea, Barents Sea, East Siberian-Laptev Seas, and Beaufort Sea (Fig. 4e). These regions also exhibit elevated surface albedo values (Fig. S4), reducing absorbed shortwave radiation and contributing to the sea ice overestimation.

Although the model generally overestimates SIT (Figs. 5 and S2), the time series analysis successfully simulates continuous thinning from ~1.8 m to ~1.3 m during 1995–2020 (Fig. 7c). Notably, however, the simulated thinning rates remain slightly lower than PIOMAS results. Stable RMSE value (~0.37) throughout this period confirm robust simulation of long-term SIT evolution. Similarly, compared to PIOMAS, the simulated SIV is consistently underestimated throughout the period, though the declining trend during 1995–2020 is well captured (Fig. 7d). For the pre-satellite era (1960–1980), evaluation using PIOMAS-20C shows E3SMv2-MPAS reproduces the 6-year cyclic "increase-decrease-increase" SIT fluctuations during 1960–1978 (Fig. 7c). While PIOMAS-20C shows no statistically significant SIT trend during 1960–1980, E3SMv2-MPAS simulates a pronounced thickening trend in this period, potentially linked to its systematic overestimation of regional ice thickness in areas like the Beaufort Sea (Fig. 5c and f, Fig. S2c). Nevertheless, across the multi-decadal scale (1960–2020), this coupled system demonstrates a reasonable representation of Arctic SIT and SIV responses to climate forcing.

#### 3.2 Surface thermohaline signatures







SST and SSS engage in complex bidirectional coupling with the atmosphere-ice system through ice/atmosphere-ocean interfacial energy-mass exchange processes. This section evaluates the spatiotemporal co-variability of SST/SSS to elucidate E3SMv2-MPAS's representation of ocean-sea ice-atmosphere interaction mechanisms.

OISST-based validation demonstrates E3SMv2-MPAS accurately reproduces key Arctic SST spatial patterns: (1) temperature gradients decreasing from shelves to central basins, and (2) warm-core features in southern Barents Sea open waters (Fig. 8a–c). Systematic regional biases are identified: the cold biases in the Greenland Sea ( $\Delta$ SST $\approx$ -2-0°C) spatially correlate with an overestimation of SIC in the same region, while positive deviations ( $\Delta$ SST $\approx$ 0-2°C) occur near Svalbard's

western coast and the Eurasian continental margins. Notably, continental coastal biases are spatially decoupled from Atlantic inflow pathways, with formation mechanisms likely associated with inaccurate vertical mixing processes stemming from stratification stability biases in shelf regions. E3SMv2-MPAS successfully captures Arctic SST warming trends during the 1995–2020 period, showing high consistency with OISST in accelerated trend characteristics (Fig. 8d). Seasonal cycle and interannual variability simulations remain within acceptable error ranges (RMSE=0.24), confirming appropriate responses to surface thermal forcing. Furthermore, the model accurately captures both the pronounced SST increase and accelerated decadal warming trend during 1995–2020 relative to the 1960–1980 baseline period. These simulated changes show a strong coupling with the accelerated decline in SIC, SIE, SIT, and SIV concurrently (Fig. 7).




Figure 8. (a–b) 1995–2020 climatological mean SST spatial distributions: (a) E3SMv2-MPAS, (b) OISST. (c) SST bias field: E3SMv2-MPAS vs. OISST. (d) Pan-Arctic (70° N–90° N) mean SST time series for 1960–1980 and 1995–2020, with dashed lines indicating linear trends (E3SMv2-MPAS: black; OISST: red) derived from least-squares regression.

E3SMv2-MPAS demonstrates comparatively weaker performance in SSS simulation versus sea ice and SST variables. Spatially heterogeneous biases are observed: negative deviations (ΔSSS=-0-1 PSU) in the Barents and Greenland Seas contrast with pronounced positive biases (ΔSSS=2-5 PSU) in the Beaufort Sea and the Kara-Beaufort shelf regions (Fig. 9a-c). The 3 PSU overestimation in the Beaufort Sea aligns with advanced assimilation model (such as HYCOM and

GLORYS12) biases reported by Hall et al. (2021), suggesting common limitations in Arctic shelf freshwater transport representation. Specifically, inadequate parameterization of surface freshwater budgets and associated processes (e.g., precipitation-evaporation fluxes, river discharge, and ice-ocean interactions) may constrain freshwater cycle simulations (Wang et al., 2024). The Beaufort Sea SIT overestimation identified previously (Fig. 5) potentially exacerbates salinity biases through reduced freshwater release (Kelly et al., 2019). If the intensity of the Beaufort Gyre is overestimated (as discussed in Section 3.1), enhanced freshwater retention could impede westward shelf transport to the Kara Sea, potentially driving salinity overestimation in the Kara-Beaufort shelf. Despite spatial biases, E3SMv2-MPAS demonstrates credible simulation of seasonal cycle phasing and amplitude in the Barents Sea SSS, while the temporal variations in the Beaufort Sea show agreement levels comparable to mainstream reanalysis products (Fig. 9d–e; Hall et al. (2021)).




Figure 9. (a-b) September 2011-December 2020 climatological mean SSS spatial distributions: (a) E3SMv2-MPAS, (b) OISSS. (c) SSS bias field: E3SMv2-MPAS vs. OISSS. (d-e) Regional SSS time series in (d) the Barents Sea and (e) the Beaufort Sea (black boxes in a-c; E3SMv2-MPAS: black; OISSS: red).

In the Greenland and Barents Seas, systematic underestimation of SST and SSS (Figs. 8c and 9c) coincides with overestimation of SIC (Figs. 3 and 4). These regions are situated within the marginal ice zone, where strong surface wind stress facilitates the transfer of energy to deeper ocean layers through the excitation of near-inertial oscillations and associated turbulent mixing processes (D'Asaro, 1985). This discrepancy may be attributed to the model's potential

overestimation of this downward energy transfer. Similarly, Zhu et al. (2022) reported that in the equatorial Pacific cold tongue region, the KPP scheme overestimates downward turbulent heat flux, leading to a cold bias in both upper-ocean and sea surface temperatures. A primary reason for these biases lies in the scheme's reliance on a single Richardson number (Ri) relationship for parameterization. Although this approach captures instability conditions in stratified shear flows, it is insufficient to uniquely determine turbulent states and mixing intensities (Zhu et al., 2022), thus limiting its performance in complex dynamic environments.

#### 3.3 Three-dimensional thermohaline structure



Accurate simulation of three-dimensional thermohaline fields remains a core technical challenge in ocean model development, directly determining model capability in representing Arctic multi-sphere coupling processes (ocean-ice-atmosphere). While preliminary evaluations of key sea ice properties (including concentration, extent, thickness, and volume)and surface thermohaline diagnostics have validated E3SMv2-MPAS's capacity to simulate Arctic shallow-layer thermal states, subsurface-to-deep thermohaline structure biases may still induce circulation distortions, material transport deviations, cross-basin exchange inaccuracies, and climate feedback misrepresentations. A multi-dimensional verification framework including spatial heterogeneity diagnostics, temporal evolution analysis and three-dimensional dynamical validation is established to assess E3SMv2-MPAS's three-dimensional thermohaline simulation performance comprehensively.

Using the 1995–2014 climatological mean profiles, systematic comparisons are conducted between E3SMv2-MPAS and observational data (Muilwijk et al., 2023) across four regions: the western Eurasian Basin, the eastern Eurasian Basin, the Chukchi Sea, and the Beaufort Sea. Thermohaline profile characteristics (0–1000 m depth) are evaluated through vertical structure evolution and regional variability analyses.

Observational data reveal maximum temperatures (1.6°C) at 250 m depth in the western Eurasian Basin, decreasing to 0°C at 800 m (Fig. 10a). E3SMv2-MPAS can successfully reproduce observed vertical temperature structure, matching the observed 250 m temperature maximum depth and maintaining temperature decline to 0°C at 1000 m depth. Despite a ∼1°C core temperature overestimation and 200 m layer thickness bias, its temperature profile RMSE is 0.448.

Figure 10. (a-d) 1995–2014 climatological mean temperature profiles from observations (Muilwijk et al., 2023), and E3SMv2-MPAS. (e-h) The same as panels (a-d) but for salinity profiles. Basins: the western Eurasian (WEB; a/e), the eastern Eurasian (EEB; b/f), the Chukchi Sea (c/g), the Beaufort Sea (d/h).



Observational spatial heterogeneity shows progressive temperature core reductions (1.6°C→1.4°C→0.8°C→0.7°C) and deepening core depths (250 m→290 m→400 m→420 m) from the western Eurasian Basin to the Beaufort Sea (Fig. 10a–d). E3SMv2-MPAS maintains systematic temperature overestimation (~1°C in the western Eurasian Basin, ~0.3°C in the Chukchi Sea and the Beaufort Sea) while successfully reproducing spatiotemporal evolution of vertical thermal structures. In salinity simulations, E3SMv2-MPAS demonstrates optimal salinity profile fitting capability through observational agreement starting from 200–300 m depth, as evidenced by the western Eurasian Basin RMSE of 0.204.

In order to systematically assess model capabilities in representing multi-scale Arctic thermal variations, an inter-decadal three-dimensional thermohaline evolution framework is established. Depth-time section comparisons between E3SMv2-MPAS and EN.4.2.2 (Good et al., 2013) are conducted to analyze spatiotemporal heterogeneity in Arctic oceanic thermal structures (Fig. 11).

Figure 11. For the Arctic Basin, (a) E3SMv2-MPAS simulated temperature profiles (0–1000 m): 1960–1980 climatology (dashed) vs. 2000–2020 climatology (solid). Right: Hovmöller diagram of depth-time evolution (1960–1980 and 2000–2020). (b) The same as panel (a) but for EN.4.2.2. (c) The same as panel (a) but for ESSMv2-MPAS minus EN.4.2.2 differences.

E3SMv2-MPAS successfully reproduces the solar radiation-driven seasonal thermal cycle observed in EN.4.2.2 (Fig. 11). Monthly thermohaline profiles (depth-month coordinates) in the upper 500 m of the Eurasian Basin better illustrate radiation-dominated seasonal characteristics: summer (June–August) surface temperature peaks coincide with salinity minima from meltwater inputs, while winter (December–February) shows sub-freezing temperatures (

Figure 12. (a-b) E3SMv2-MPAS simulated 1995-2020 climatological mean (left) temperature and (right) salinity profiles (0-500 m) in the Eurasian Basin (EB), with Hovmöller diagrams of monthly variability. (c-d) The same as panels (a-b) but for WOA23. (e-f) The same as panels (a-b) but for model-observation differences (E3SMv2-MPAS minus WOA23).

E3SMv2-MPAS demonstrates exceptional multi-temporal simulation capabilities for AW dynamics (Fig. 11). Observations reveal stable AW core temperatures (~1.6°C) during 1960–1980, increasing to ~2°C in 2000–2020 with core shallowing from 350 m to 300 m in the whole Arctic Basin (Fig. 11b). E3SMv2-MPAS accurately reproduces both the ~0.4°C warming magnitude and ~50 m vertical migration (Fig. 11a). However, regional-specific biases emerge in seasonal variability simulations (Fig. 12). EN.4.2.2 identifies semi-annual signals in the 200–500 m layer of the Eurasian Basin (September–November peaks at ~1.5°C; Fig. 12c), linked to winter Atlantification intensification. E3SMv2-MPAS fails to capture this seasonality, producing persistent warm biases in 200–400 m layers with overestimated spring–summer core temperatures (0.5–0.8°C; Fig. 12e). This discrepancy may be attributed to the GM parameterization scheme, which models mesoscale eddy effects on heat and salt redistribution through bolus advection and Redi diffusion. In general, the Arctic winter features greater mixed layer depth and weaker stratification due to brine rejection during sea ice formation and wind-driven stirring (Peralta-Ferriz and Woodgate, 2015). These processes promote eddy penetration, increasing the efficiency of vertical heat transport. In contrast, strengthened stratification in summer restricts the vertical scale of eddies and reduces heat transfer. However, the GM scheme employs a fixed diffusion coefficient, which prevents it from capturing the seasonal variability modulated by stratification changes.

We further investigate the decadal-scale thermohaline variability across Arctic basins. The results reveal regional heterogeneity in temperature and salinity trends (Fig. 13). Inter-decadal comparisons (1970s vs. 2000s–2020s; Muilwijk et al. (2023)) reveal pan-Arctic synchronous warming across the Eurasian Basin sectors and the Amerasian sub-regions (Fig. 13e–h). However, E3SMv2-MPAS underestimates the warming in the Amerasian Basin (0.1–0.5°C biases; Fig. 13c–d and g–h).

Figure 13. (a-d) Vertical profiles of climatological mean temperature (red curves) and salinity (blue curves) in the western Eurasian Basin (WEB; a), the eastern Eurasian Basin (EEB; b), the Chukchi Sea (c), and the Beaufort Gyre (d) from E3SMv2-MPAS: dashed lines denote 1970s (1971–1979), solid lines represent 2000s–2010s (2001–2019). (e-f) Corresponding observational profiles from Muilwijk et al. (2023) with identical temporal averaging.



In the Eurasian Basin upper layers (~100–450 m; Fig. 13a–b and e–f), observations show dual-mode thermal evolution: shallow warming above temperature cores (100–250 m) contrasts with systematic warming below (250–450 m). Model simulations exhibit spatial heterogeneity: 0.2±0.1°C underestimation of shallow warming contrasts with excessive vertical response ranges (250–1000 m vs. observed 250–450 m). Notably, simulated AW layer thickening in the eastern Eurasian Basin during 2000s–2010s lacks observational support (Fig. 13b and f). These discrepancies may be partly attributed to

biases in the representation of vertical processes. As indicated by sensitivity experiments such as those of Liang & Losch (2018), enhanced vertical mixing could promote upward heat transport from AW, potentially causing cooling at intermediate depths (200–900 m). Our model uses a relatively low background diffusivity (1.0×10<sup>-5</sup> m<sup>2</sup> s<sup>-1</sup>), which remains constant across time periods despite evidence that Arctic amplification and Atlantification in the 2000s–2010s (Polyakov et al., 2017, 2025; Rantanen et al., 2022; Richards et al., 2022; Shu et al., 2022) may have strengthened vertical mixing compared to the 1970s. The model's failure to represent this temporal increase in mixing efficiency might have limited upward heat transfer, confining warming mainly to intermediate and deeper layers – consistent with the underestimation of shallow warming and exaggerated deep response seen in our simulations.





In the Chukchi Sea, observations indicate basin-wide warming from core layers to AW bottom (~1000 m), showing  $\Delta T = 0.2 \pm 0.1$ °C (Fig. 13c-d and g-h). While successfully reproducing Chukchi thermal trends, the model exhibits systematic Beaufort Sea deviations. Salinity changes primarily occur in the upper 300 m of the Amerasian Basin (the Chukchi and Beaufort Seas; observed  $\Delta S=-0.3\pm0.2$  PSU), with the model failures in capturing the freshening of the Chukchi Sea and underestimation of trends in the shallow-layer (<80 m) of the Beaufort Sea. The simulated salinity biases may be related to the use of an inappropriately high and constant isopycnal diffusion coefficient (κ=300 m<sup>2</sup> s<sup>-1</sup>) in the GM parameterization. 495 This high diffusion coefficient likely results in excessively strong along-isopycnal mixing, which oversmooths horizontal salinity gradient fronts formed by freshwater accumulation (e.g., from melting ice and increased runoff). During the 1970s, when background freshwater signals were relatively weak, the effect of strong diffusion was less pronounced. However, under the strongly increased freshwater input in the 2000s–2010 (Polyakov et al., 2013; Wang et al., 2019), the persistently 500 high κ value continuously and excessively diffused the simulated upper-layer low-salinity anomalies, hindering their realistic accumulation and maintenance in the basin upper layer. As a result, the model significantly underestimates the magnitude of decadal freshening observed in the region.

AW demonstrates systematic cooling and freshening (temperature and salinity reduction) during its transport from the Eurasian to the Amerasian Basin (Fig. 10), a transformation likely modulated by baroclinic adjustment processes in the interbasin transition zone. These processes govern cross-basin material-energy exchange (Aksenov et al., 2016). We analyze coordinated meridional sections along 145° W in the Amerasian Basin and 70° E in the Eurasian Basin to access variability in AW properties across space (Fig. 14). WOA23-based comparisons confirm E3SMv2-MPAS's capability in reproducing inter-basin gradient characteristics through three key aspects: (1) AW thermal attenuation: Successful simulation of core temperature decreases from the Eurasian to the Amerasian Basin, replicating thermodynamic dissipation processes; (2) Stratification depth displacement: Realistic representation of westward-decreasing upper boundary depths matching slope current adjustments; (3) Surface freshwater transport effects: Accurate reproduction of the surface salinity depression in the Amerasian Basin relative to the Eurasian Basin, validating appropriate parameterization of Pacific-origin freshwater influx mechanisms. Persistent thermal biases in the Eurasian Basin emerge in 145° W sections, with maximum +2°C warm deviations in 100–500 m core layers (Fig. 14e). Despite absolute temperature biases, maintained meridional heat transport gradients confirm fundamental physical framework validity for large-scale advection processes.

Figure 14. (a-b) E3SMv2-MPAS simulated 1995–2020 climatological (left) temperature and (right) salinity distributions along the 145° W-70° E transect (location mapped in Fig. 1b). (c-d) The same as panels (a-b) but for WOA23. (e-f) The same as panels (a-b) but for the model-observation differences (E3SMv2-MPAS minus WOA23).

#### 3.4 Freshwater content spatiotemporal variability




The Arctic Ocean constitutes a major freshwater reservoir within the global climate system. Since the mid-1990s, the freshwater content (FWC) in the Arctic Ocean has exhibited a marked increasing trend, primarily driven by persistent anticyclonic atmospheric forcing over the Beaufort Gyre region (1997–2018) and the reduction of Arctic sea ice (Proshutinsky et al., 2019; Wang et al., 2024). This excess freshwater is exported into the North Atlantic through Fram Strait and Davis Strait (Wang et al., 2019), eventually reaching convection regions in the Labrador and Greenland–Iceland–Norwegian Seas. These areas are critical for the formation of global deep waters, which act as a key driver of large-scale ocean circulation systems, including the Atlantic Meridional Overturning Circulation (AMOC) (Arzel et al., 2007). The southward transport of freshwater may influence circulation dynamics by reducing seawater density and suppressing vertical mixing (Haine et al., 2023). Furthermore, Arctic freshwater variability significantly influences ecosystem structure and function (Proshutinsky et al., 2019). Therefore, accurately assessing the freshwater content in the Arctic Ocean represents a central challenge in physical oceanography and climate dynamics, with major implications for understanding both climate variability and long-term change (Haine et al., 2023).

The FWC is defined as follows (Wang et al., 2024):



$$FWC = \int_{H}^{0} \frac{S_{ref} - S}{S_{ref}} dz, \tag{1}$$

where S denotes salinity,  $S_{ref}$  is the reference salinity – set to 34.8 psu, the mean Arctic Ocean salinity according to Aagaard and Carmack (1989) – and H represents the depth at which salinity equals  $S_{ref}$ .

Based on this formulation, we evaluate the spatial distribution of the multi-year mean (1995–2020; Fig. 15a–b) and decadal differences (2005–2014 vs. 1995–2004; Fig. 15c–d) of FWC as simulated by E3SMv2-MPAS, in comparison with observational data from WOA23. Additionally, we analyze the basin-wide averaged time series of FWC across the Arctic Ocean (Fig. 15e).

Figure 15. Spatial distribution and temporal evolution of FWC (in meters) in the Arctic Ocean. (a-b) Multi-year mean (1995–2020) FWC from (a) E3SMv2-MPAS and (b) WOA23. (c-d) Decadal difference in FWC (2005–2014 minus 1995–2004) from (c) E3SMv2-MPAS and (d) WOA23. (e) Time series of the Arctic Ocean-wide averaged FWC from E3SMv2-MPAS for the period 1995–2020.

As shown in Fig. 15a-b, E3SMv2-MPAS generally captures the spatial characteristics of Arctic FWC, such as the increase from the Eurasian Basin toward the Amerasian Basin and the maximum values located in the Beaufort Sea. However, the model overestimates FWC in the vicinity of Baffin Bay.

Observational results from WOA23 indicate a pronounced strengthening of FWC in the Beaufort Sea during 2005–2014 compared to 1995–2004. The model successfully reproduces this decadal change in that area (Fig. 15c–d). Nonetheless, E3SMv2-MPAS erroneously simulates a significant decrease in FWC across the Eurasian and Makarov Basins, where WOA23 shows a slight increase. Moreover, the model overestimates the increase in FWC along a pathway extending from the east of Greenland to the northern Canadian Archipelago and into the Canada Basin.

The time series of the total Arctic FWC (Fig. 15e) exhibits a fluctuating upward trend from 1995 to 2020. Polyakov et al. (2013) reported a notable acceleration in Arctic FWC accumulation during the 2000s, particularly between 2003 and 2010 – a trend that aligns with the sharp rise simulated by E3SMv2-MPAS between 2002 and 2008. Furthermore, Wang et al. (2019) noted a levelling off of the FWC growth trend after 2010, which is also captured by the model.

## 3.5 Gateway transports: volume, heat, and freshwater







Over the past decades, the Arctic climate system has undergone rapid changes, including shifts in sea ice, atmosphere, and ocean conditions (Landrum and Holland, 2020; Polyakov et al., 2005; Shu et al., 2022). These rapid changes are closely linked to the lateral exchanges of heat and freshwater across the boundaries of the Arctic (Von Schuckmann et al., 2020). The Arctic Ocean's connections to other oceans are defined by four major gateways (from east to west): Bering Strait, Fram Strait, Barents Sea Opening, and Davis Strait (Tsubouchi et al., 2024). These critical gateways not – as discussed in the previous section – serve as major pathways for freshwater export from the Arctic, which in turn influences global deepwater formation, large-scale circulation, and ecosystems, but also subject Arctic sea ice and ocean conditions to strong influences from Atlantic and Pacific water inflows. These impacts include modulating sea ice cover (Årthun et al., 2012, 2019; Docquier and Koenigk, 2021), ocean stratification (Veneziani et al., 2022), ecosystem (Woodgate and Peralta-Ferriz, 2021), ocean temperature (Barton et al., 2018), and freshwater content (Woodgate, 2018). Therefore, the accurate simulation of volume, heat, and freshwater transports through these four major gateways – including their interannual and decadal variability – is crucial. In this section, we evaluate the performance of the E3SMv2-MPAS in simulating these key exchanges through comparison with multi-source observational data.

The oceanic net volume transport (VT), heat transport (HT), and freshwater transport (FWT) through the key gateways—Bering Strait, Fram Strait, Barents Sea Opening, and Davis Strait—are calculated as follows (Karpouzoglou et al., 2022; Shu et al., 2022; Wang et al., 2024):

$$VT = \int_{-H(\lambda)}^{0} \int_{\lambda_1(z)}^{\lambda_2(z)} V d\lambda dz, \qquad (2)$$

$$HT = \rho_o c_p \int_{-H(\lambda)}^0 \int_{\lambda_1(z)}^{\lambda_2(z)} V(T - T_{ref}) d\lambda dz, \tag{3}$$

$$FWT = \int_{-H(\lambda)}^{0} \int_{\lambda_1(z)}^{\lambda_2(z)} V \frac{(S_{ref} - S)}{S_{ref}} d\lambda dz, \tag{4}$$

Here, V, T, and S denote velocity, potential temperature, and salinity, respectively;  $\rho_o$  is the seawater density; and  $c_p$  represents the specific heat capacity of seawater. The reference temperature  $T_{ref}$  is set to 0 °C, and the reference salinity  $S_{ref}$  is defined as 34.8, corresponding to the mean salinity of the Arctic Ocean (Aagaard and Carmack, 1989). The integration is performed over the full depth H – defined as the bathymetry along the transect – and across the lateral extent  $\lambda$  of each strait. Units for VT, HT, and FWT are Sverdrup (Sv; 1 Sv=10<sup>6</sup> m<sup>3</sup> s<sup>-1</sup>), Terawatt (TW), and km<sup>3</sup> year<sup>-1</sup>, respectively.

## 3.5.1 Bering Strait




Observational data from the Bering Strait between 2000 and 2018 (Woodgate and Peralta-Ferriz, 2021) reveal significant increasing trends in volume, heat, and freshwater transports (Fig. 16a–c). However, E3SMv2-MPAS fails to reproduce these overall trends. The discrepancies in volume and freshwater transports may be attributed to biases in the JRA55 reanalysis and river runoff forcing data (Wang et al., 2024). Although the model does not capture the increasing trends in volume and freshwater transports during 2000–2012, it successfully simulates the upward trends from 2012 to 2018, including interannual variability, with deviations generally within 0.2 Sv and 500 km³ year⁻¹ (Fig. 16a and c). The similar interannual and decadal variability between simulated volume and freshwater transports indicates that the model accurately represents the mechanism whereby increased freshwater transport is primarily driven by volume transport (Woodgate and Peralta-Ferriz, 2021). For heat transport, the model also captures the rapid increasing trend observed after 2012 (Fig. 16b).

Figure 16. Time series of net (a, d, g, j) volume, (b, e, h, k) heat, and (c, f, i, l) freshwater transport through the (a-c) Bering Strait (BS), (d-f) Fram Strait (FS), (g-i) Barents Sea Opening (BSO), and (j-l) Davis Strait (DS) from 1995 to 2020. Black and red lines denote simulated results from E3SMv2-MPAS and observational estimates, respectively. Positive values represent transport into the Arctic Ocean, while negative values denote transport out of the Arctic Ocean. Observational data are from Woodgate and Peralta-Ferriz (2021) for BS; Tsubouchi et al. (2024) for volume and heat transport in FS; Karpouzoglou et al. (2022) for freshwater transport in FS; and Tsubouchi et al. (2024) for BSO and DS. The locations of the four straits are highlighted in green in Fig. 1b.

#### 3.5.2 Fram Strait


A study by Schauer et al. (2004) shows annual mean net volume transport through the Fram Strait between -4±2 Sv and -2±2 Sv during 1997–2000. Schauer et al. (2008) further indicate values of -2±5.9 Sv (1997–2002) and -2±2.7 Sv (2002–2006). Despite considerable uncertainties in observations, these estimates confirm that the simulated volume transport by E3SMv2-MPAS falls within a plausible range (Fig. 16d). Results from Tsubouchi et al. (2024) for 2005–2009 show that, although the model generally overestimates southward volume transport, it reproduces the interannual variability reasonably well (Fig.

16d). Observations indicate that the annual mean net heat transport increased from 16±12 TW in 1997 to 41±5 TW in 1999 (Schauer et al., 2004). In comparison, the model overestimates heat transport in 1997 but accurately captures both the pronounced increasing trend during 1997–1999 and the value in 1999 (Fig. 16e). Compared to observational data from 2005–2009 (Tsubouchi et al., 2024), the simulated heat transport values agree well in magnitude, and the model largely reproduces the initial decrease followed by an increase during this period. Moreover, the model successfully captures both the increasing trend and the magnitude of the observed southward freshwater transport through the Fram Strait from 2004 to 2017 (Karpouzoglou et al. (2022); Fig. 16f).

## 3.5.3 Barents Sea Opening


Between 2000 and 2009, the annual mean net volume and heat transports through the Barents Sea Opening were reported as 2.3 Sv and 70±5 TW, respectively (Smedsrud et al., 2013), which are generally consistent with the model simulations (Fig. 16g-h). E3SMv2-MPAS also successfully reproduces the decreasing trends in both volume and heat transport during 2005–2008, albeit with slight systematic overestimation. Observations indicate a pronounced decreasing trend in freshwater export through the Barents Sea Opening during 2005–2009 (Tsubouchi et al., 2024). While the model captures this trend, it overestimates the magnitude (Fig. 16i). Additionally, the mean freshwater transport through the Barents Sea Opening between 2000 and 2010 was -90±90 km³ year¹ (Haine et al., 2015), further supporting the model's tendency to overestimate freshwater export in this region.

#### 3.5.4 Davis Strait

Observations from 2004 to 2010 report annual mean net volume and freshwater transports through the Davis Strait as - 1.6±0.5 Sv and -2,900±190 km³ year⁻¹, respectively (Curry et al., 2014). E3SMv2-MPAS simulations agree well with these values during the same period (Fig. 16j and l). The model accurately captures the increased freshwater export through the Davis Strait in the mid-to-late 2010s (particularly 2015–2017), which is influenced by Arctic-external atmospheric forcing affecting sea level variability (Wang et al. (2022); Fig. 16l). According to Wang et al. (2022), the freshwater export through the Davis Strait increased by over 1500 km³ year⁻¹ between 2010 and 2017, a magnitude quantitatively reproduced by E3SMv2-MPAS. However, the simulated heat transport ranges from 13 to 19 TW during 2005–2009, underestimating the observed range of 19–27 TW (Tsubouchi et al. (2024); Fig. 16k).

## 4 Atlantic Water layer states

#### 4.1 Parametric characterization of Atlantic Water core

As demonstrated in Section 3, model biases predominantly manifest in two critical parameters: AW core temperature (AWCT) and depth (AWCD). These metrics, defined as the maximum temperature within 150–900 m depth and its corresponding depth (Khosravi et al., 2022; Shu et al., 2022; Wang et al., 2024), are employed to evaluate E3SMv2-MPAS's

performance in reproducing spatiotemporal features of AW (Fig. 17). Observational AWCT/AWCD datasets from Richards et al. (2022) reveal successful model reproduction of baseline spatial gradients: decreasing AWCT and increasing AWCD from the Eurasian to the Amerasian Basin, though with marked regional heterogeneity (Fig. 17a–d). Systematic overestimation of AWCT ( $\pm$ 0.5°C) is identified in the western Eurasian Basin off-shelf regions (high-latitude sectors), while negative deviations ( $\pm$ 0.5°C) occur in the Beaufort Sea. AWCD simulations demonstrate higher accuracy, with minor underestimation ( $\pm$ 0.5°C) m) in the eastern Eurasian Basin.


Figure 17. (a-b) 1995-2018 climatological mean AWCT spatial distributions: (a) E3SMv2-MPAS vs. (b) observation from Richards et al. (2022). (c-d) The same as panels (a-b) but for AWCD. (e-f) Temporal evolution of basin-averaged AWCT (top row) and AWCD (bottom row) in the Eurasian Basin (e) and the Amerasian Basin (f): E3SMv2-MPAS (black) versus observations (red).

Interannual variability (1995–2018) is adequately captured through basin-averaged AWCT/AWCD magnitudes (Fig. 17e–f). However, post-2013 increases in AWCD in the Amerasian Basin remain unresolved. While demonstrating credibility in long-term trend simulations, model responsiveness to decadal-scale climatic shifts requires further optimization – critical for predicting nonlinear Atlantification trajectories.

To address the systematic underestimation of Atlantification in model simulations (mentioned in Section 1), five key parameters are quantified: AWCT, AWCD, AW upper boundary (0°C isotherm; Meyer et al. (2017)), AW layer thickness (between 0°C isotherms), and AW heat content. By analyzing their spatiotemporal response characteristics, this study investigates the trans-decadal evolution of Atlantification.

The AW heat content is calculated as follows (Polyakov et al., 2017):






advective-diffusive redistribution.

$$Q = \int_{z_1}^{z_2} \rho_w \, c_p \left(\theta - \theta_{freezing}\right) dz,\tag{5}$$

where  $z_1/z_2$  denote layer boundaries,  $\rho_w$  seawater density,  $c_p$  specific heat of seawater, and  $\theta_{freezing}$  freezing temperature. Both basins exhibit coordinated changes during 1960s–1980s and 2000s–2020s: AWCT increases, AWCD decreases, AW upper boundary shallows, layer thickness expands, and heat content accumulates (Fig. 18). Post-2000s acceleration of these trends shows tight coupling with enhanced Atlantic meridional heat transport under Arctic amplification. E3SMv2-MPAS captures the key thermodynamic signatures of Atlantification (the Eurasian Basin vs. the Amerasian Basins between 2000–2020), aligning closely with observationally derived mechanisms of AW intrusion and its climatic impacts (Polyakov et al., 2017): (1) A 1°C gradient in AWCT between the Eurasian and Amerasian Basins, consistent with zonal heat dissipation; (2) A 130-m shallower AWCD in the Eurasian Basin, reflecting intensified vertical mixing due to sea ice loss; (3) Synergistic changes in AW layer thickness and heat content (100 m thinner layer with +4000 MJ·m<sup>-2</sup> in the Eurasian Basin), confirming

Figure 18. (a-b) 1960–1980 vs. 2000–2020 climatological mean AWCT in the (a) Eurasian basin (EB) and (b) Amerasian Basin (AAB). (c-j) The same as panels (a-b) but for AWCD (c-d), AW layer upper boundary depth (AWupdepth; e-f), thickness (AWthickness; g-h), and heat content (Q\_AW; i-j).

This multi-scale validation confirms E3SMv2-MPAS's physical credibility in reproducing Atlantification mechanisms: cascading heat flux-stratification-heat content responses and inter-basin thermodynamic evolution. The model thus provides critical process fidelity for predicting Arctic oceanic thermal threshold transitions.

## 675 4.2 Coupling between the Atlantic Water and surface layers






The AW layer constitutes the most critical oceanic heat reservoir in the Arctic Ocean (Carmack et al., 2015), containing sufficient thermal energy to melt all Arctic sea ice within several years (Turner, 2010) and capable of dissolving 3–4 times the current ice volume (Carmack et al., 2015; Polyakov et al., 2020b). A pronounced halocline characterized by rapidly increasing salinity with depth typically separates the cold, low-salinity surface waters from the warm, saline AW in the Eurasian and Amerasian Basins. This strong stratification effectively inhibits vertical water mass exchange (Peralta-Ferriz and Woodgate, 2015), isolating the AW layer from sea ice and mixed layer interactions (Aagaard et al., 1981; Richards et al., 2022). Under these physical constraints, vertical heat transport primarily occurs through molecular-scale processes involving internal wave breaking and double-diffusive mixing (Davis et al., 2016). However, since the 1970s, progressive weakening of the eastern Eurasian Basin halocline has been documented (Polyakov et al., 2010; Steele and Boyd, 1998), culminating in its complete failure as an effective thermal barrier for intermediate AW heat by the mid-2010s (Polyakov et al., 2020a). Stratification collapse has triggered a regime shift from double-diffusive dominance to shear-driven turbulent mixing, fundamentally altering vertical heat flux dynamics (Polyakov et al., 2020a).

The KPP scheme employed by E3SMv2-MPAS driven by Gradient Ri physics (Zhu et al., 2022). This study evaluates whether this parameterization scheme, combined with the model's unstructured mesh capability, adequately resolves Arctic vertical thermal coupling features, particularly in the Eurasian Basin. A diagnostic framework based on spatiotemporal correlation analysis is established to quantify the thermal linkage between the upper (10 m) and intermediate (AW core layer, 400 m) ocean layers. This analysis addresses two critical aspects: (1) spatiotemporal delay characteristics in vertical heat signal propagation relative to AW transport timescales, and (2) potential regime shifts in interlayer coupling mechanisms under climate warming. This diagnostic framework provides dynamic constraints for optimizing vertical mixing parameterizations while elucidating climate impacts of upper-ocean thermal variability.

During 1960–1980 baseline conditions (zero time lag), statistically significant positive correlations (p

Figure 19. (a) 1960–1980 climatological mean correlation between surface (5 m) and mid-depth (400 m) temperatures. (b-h) Lagged correlations at 6-month intervals (lag 6 mons to 42 mons). Black dots indicate significance (p

Figure 20. The same as Fig. 19, but for 1995–2020 period.

A fundamental regime shift in Arctic intermediate-to-surface thermal coupling mechanisms under climate warming is revealed through cross-temporal-scale lagged correlation diagnostics: transitioning from historical basin-scale inertial transport patterns to contemporary instantaneous response modes. This regime shift, driven by altered AW thermohaline properties and reduced stratification stability, enhances vertical heat leakage efficiency from intermediate layers. Model evaluation demonstrates that while the KPP scheme captures accelerated heat transport trends, systematic biases persist in nonlinear responses to shear mixing (Figs. 19 and 20). Future research directions emphasize developing scale-aware parameterizations incorporating high-resolution turbulence observations to improve model capabilities in predicting Arctic energy transport regime shifts.

#### 5 Discussion





## 5.1 Comparison with OMIP2 models under diverse grid configurations and resolutions

To evaluate the performance of different ocean-sea ice coupled models in simulating the three-dimensional themohaline structure, particularly that of the intermediate AW layer, we further discuss five resolution-matched model pairs from OMIP2 (Wang et al., 2024). Thermohaline profile characteristics in the Eurasian and Amerasian Basins are systematically compared between high/low-resolution model pairs (solid/dashed lines), E3SMv2-MPAS, and WOA23 data (Locarnini et al., 2024; Reagan et al., 2024) to elucidate ocean model grid configuration impacts (Fig. 21).

Low-resolution models exhibit systematic biases as follows: (1) Substantial underestimation of AWCT (e.g., 

Figure 21. (a-b) 1995-2014 climatological mean temperature profiles in (a) the Eurasian Basin and (b) the Amerasian Basin: Observations (WOA23; red), E3SMv2-MPAS (black), OMIP2 models (Wang et al. (2024); dashed: low-resolution, solid: high-resolution). (c-d) The same as panels (a-b) but for salinity profiles.

Following the evaluation of the three-dimensional thermohaline structure simulations in the OMIP2 models, we further examine their capability to reproduce the spatial distribution of AWCT. As indicated by cross-validation within the OMIP2 framework, among the five resolution-varied model groups, only FESOM\_4.5km, MOM\_3.6km, and HYCOM\_3.6km demonstrate high AWCT spatial pattern simulation skills (Fig. 22). FESOM\_4.5km outperforms E3SMv2-MPAS (10 km) in representing the western Eurasian Basin shelf-basin gradients, but underperforms in the Amerasian Basin (Fig. 22b–c). Low-resolution models exhibit a systematic underestimation of AWCT, with FESOM\_24km being the exception, reaffirming unstructured meshes' polar ocean modeling advantages (Fig. 22h–l).

Figure 22. 1995–2018 climatological mean AWCT spatial patterns from (a) observations (Richards et al., 2022), (b) E3SMv2–MPAS, and (c–l) OMIP2 models (Wang et al., 2024). Middle/bottom rows: High-resolution and corresponding low-resolution model pairs from OMIP2.

Despite comparable resolutions to other high-resolution models (e.g., ACCESS-MOM 9 km, FSU-HYCOM 32 km), unstructured mesh configurations enable refined representation of key hydrographic gateways like the Fram Strait. Compared to tripolar grid models suffering numerical dissipation near complex coastlines, variable mesh designs achieve reduced the Eurasian Basin temperature errors under equivalent computational resources. Model grid type and computational efficiency exhibit nonlinear relationships. Unstructured meshes (FESOM/MPAS) permit dynamic optimization through localized refinement in critical regions (e.g., AW intrusion pathways). This targeted refinement strategy provides new technical approaches for Arctic ocean modeling, particularly under accelerating Atlantification processes.

# 5.2 Sources of systematic biases and trade-offs between resolution and parameterizations

Analyses in Section 3 not only discussed the simulation biases of E3SMv2-MPAS but also traced their potential origins. For most biases, the primary causes can be attributed to inadequacies in physical parameterizations. First, the inadequate representation of eddy dynamics is a key source. For instance, the underestimation of freshening in the Amerasian Basin may result from the use of a fixed eddy diffusivity ( $\kappa$ =300 m<sup>2</sup> s<sup>-1</sup> in the Arctic), which oversmooths salinity fronts. Similarly, the model's failure to capture the seasonal variability of the AW layer likely stems from the invariant  $\kappa$  in the GM scheme, which cannot respond to the seasonal cycle of sea ice retreat and associated changes in stratification. Second, limitations in vertical mixing parameterizations act as another key source. The coordinated biases in SST, SSS, and SIC in the Greenland and Barents Seas, for example, may arise from the inherent limitations of the KPP scheme's single Ri-based approach in defining turbulent states and mixing intensities within complex dynamic environments. Additionally, the misrepresentation of the warming layer in the Eurasian Basin could be linked to inappropriate background diffusion coefficients within the KPP framework.

Increasing model resolution presents an effective pathway to reduce reliance on empirical parameterizations by more directly resolving key physical processes, such as mesoscale eddies. Enhanced resolution can, to some extent, mitigate the inaccuracies of existing schemes. For instance, studies have shown that higher resolution improves the simulation of the AW layer's temperature, thickness, spatial distribution, and its decadal warming trends (Wang et al., 2024). However, the small Rossby radius of deformation (often ≤3 km) in the Arctic (Veneziani et al., 2022) implies that even with computationally feasible resolution increases, critical processes (e.g. mesoscale eddies, vertical mixing, and ice-ocean interactions) may remain under-resolved (Chassignet et al., 2020; Wang et al., 2018). Therefore, the development of more advanced physical parameterizations remains imperative. It is noteworthy that resolution increases have proven effective in improving the simulation of volume, heat, and freshwater transports through critical gateways such as the Fram Strait and Davis Strait (Wang et al., 2024). The Fram Strait, in particular, serves as a pivotal channel for Atlantic heat influx into the Arctic Ocean (Herbaut et al., 2022; Pnyushkov et al., 2021). In conclusion, we propose that a cost-effective strategy involves targetedly increasing resolution in key gateway regions while concurrently refining parameterizations for mesoscale eddies and vertical mixing.

## 5.3 Limitations of the experimental design







Due to computational resource constraints, this study adopted a two-phase simulation strategy with non-consecutive time periods: first, the model was integrated from 1958 to 1981, and the final state of this period was used as the initial condition to directly start the simulation for the 1995–2020 period. Although this approach effectively reduced computational costs, and both previous studies and our model diagnostics indicate that key upper-ocean and sea ice variables had reached a quasi-equilibrium state by 1981, skipping the continuous integration of the 1982–1994 period may introduce certain limitations. For instance, the simulation of some medium- to long-term fluctuations or memory-dependent processes might be affected.

Should computational resources allow in the future, we will perform a continuous simulation from 1958 to 2020 to more accurately reproduce the evolution of the climate system.

Furthermore, since only a single JRA55 forcing cycle was applied, the deep ocean and some physical quantities may not have fully departed from the influence of the initial PHC hydrographic fields or reached complete equilibrium. This could potentially affect the stability and initial-condition independence of the simulation results. In subsequent work, given sufficient resources, we plan to carry out at least three full JRA55 forcing cycles to promote more complete adjustment of the ocean state, reduce dependence on initial conditions, and thereby enable a more comprehensive and robust evaluation of the climate performance of the E3SMv2-MPAS.

#### 5.4 Limitations from the atmospheric forcing: the JRA55 warm bias

The JRA55 reanalysis forcing data employed in E3SMv2-MPAS exhibits a known warm bias over the central Arctic deep basin (Batrak and Müller, 2019). This bias may systematically suppress sea ice growth and induce upper-ocean warming in the simulation by enhancing downward longwave radiation and reducing oceanic sensible heat loss. Therefore, the overestimated SST (Fig. 8c) and underestimated summer SIC (Fig. 4e–g) simulated in the central Arctic basin may be partially attributable to the inherent bias in the forcing data, rather than solely to inaccuracies in the model's physical processes.

The enhanced ice melt driven by this warm bias releases additional freshwater, leading to a stronger and shallower freshwater layer (a more pronounced halocline) in the surface ocean, which significantly strengthens the stratification stability of the upper ocean. This inhibits vertical mixing between layers and impedes the upward heat transfer from the warmer, saltier AW below. This bias may partly explain the overestimation of the intermediate AW layer temperature alongside the underestimation of the mixed-layer temperature (Fig. 12e). Future work will consider employing alternative reanalysis products or applying bias-correction methods to better constrain the impact of forcing uncertainties on simulation results.

# 820 6 Conclusions




This study systematically evaluates the Arctic ocean-sea ice simulation capabilities of E3SMv2-MPAS through multi-source observations (in situ profiles, satellite remote sensing, optimum interpolation datasets) and reanalysis products (NSIDC, ERA5, HadISST1, PIOMAS, ORAS5), with focus on core parameters including ice sea (concentration/extent/thickness/volume; SIC/SIE/SIT/SIV), surface thermohaline properties (sea surface temperature/salinity; SST/SSS), three-dimensional thermohaline structures, freshwater content (FWC), gateway transports, Atlantic Water (AW) heat characteristics, and vertical thermal linkages. Spatial distribution patterns, seasonal-to-decadal variability, and three-dimensional evolutionary processes are comprehensively analyzed.

E3SMv2-MPAS demonstrates significant advantages in Arctic climatology simulations: (1) Accurate representation of spatial heterogeneity in SIC, SIT, and SST (Figs. 3–5, 8a–c); (2) Realistic simulation of interannual and decadal variability in SIC, SIE, SIT, SIV and SST, along with a highly consistent reproduction of the 1995–2020 SIC decline trend compared to NSIDC observation, outperforming Hadley and ERA5 reanalysis products (Fig. 7a); (3) Consistent SSS spatial patterns and seasonal evolution with leading reanalysis products including HYCOM and GLORYS12 (Fig. 9; Hall et al. (2021)); (4) Faithful reproduction of both the spatial distribution and long-term trend of Arctic FWC (Fig. 15); (5) Accurate simulation of volume, heat, and freshwater transports through key Arctic gateways, capturing their observed magnitudes and essential variability trends (Fig. 16).

E3SMv2-MPAS demonstrates exceptional capability in simulating the three-dimensional thermohaline structure and variability of the AW layer, accurately capturing its key thermodynamic and dynamic processes in the Arctic Ocean: (1) Precise reproduction of AW layer thickness/depth and core temperature (Figs. 10 and 17); (2) Effective capture of AW warming trends including decadal-scale intermediate layer heating and vertical shoaling of warm cores (Figs. 11 and 13); (3) Realistic simulation of accelerated Atlantification processes, evidenced by post-2000 intensification in AW core temperature and heat content while reduced AW core depth, upper boundary and layer thickness, and instantaneous surface-intermediate heat transfer in the Eurasian Basin (Figs. 18–20). Additional breakthroughs include successful representation of solar-driven seasonal upper-ocean thermal cycles (Fig. 12) and inter-basin water mass gradient evolution from the Eurasian Basin to the Amerasian Basin (e.g., AW thermohaline attenuation, vertical stratification shifts, and surface freshwater transport effects; Fig. 14). These advancements establish critical numerical platforms for investigating Arctic stratification destabilization and cross-scale energy transfer mechanisms.





Notwithstanding these achievements, key limitations persist: (1) Systematic overestimation of SIT in the Canadian Basin (0.5–1 m bias; Fig. 5); (2) Coordinated underestimation of SST/SSS and overestimation of SIC in the Greenland and Barent Seas (Figs. 3–4, 8a–c, 9a–c); (3) Residual overestimation of AW core temperature (0–1°C) and errors in seasonal Atlantification phase (Figs. 10 and 12); (4) Asymmetries in regional decadal thermohaline evolution (e.g., underestimated upper-layer warming and overestimated deep warming in the Eurasian Basin, unresolved mid-layer warming and upper-layer freshening trends in the Amerasian Basin; Fig. 13).

This study confirms that E3SMv2-MPAS significantly enhances simulation capabilities for Arctic oceanic thermal structures and cross-layer coupling processes through high-resolution unstructured meshes, establishing crucial technical references for polar climate model development in the CMIP7 era.

Code and data availability. The E3SM model code is publicly available via the https://github.com/E3SM-Project/E3SM/releases. Instructions on how to configure and execute E3SM are available at https://e3sm.org/model/runninge3sm/e3sm-quick-start/. All simulations detailed in Section 2.1 can be regenerated by executing the code hosted in this repository: https://doi.org/10.5281/zenodo.15493256 (Lv, 2025). Preprocessing of E3SMv2-MPAS outputs utilized nco-5.1.1, accessible through the https://nco.sourceforge.net/. The JRA55-v1.5 atmospheric forcing data driving the simulations were obtained from the https://aims2.llnl.gov/search/input4mips/. ETOPO 2022 bathymetry was derived from the https://www.ncei.noaa.gov/products/etopo-global-relief-model. Model evaluations employed the following observational and reanalysis products: Sea ice concentration: NSIDC (https://noaadata.apps.nsidc.org/NOAA/G02202 V4/north/aggregate/), Met Office Hadley Centre observational datasets (https://www.metoffice.gov.uk/hadobs/), and ERA5 monthly single-level data (https://cds.climate.copernicus.eu/datasets/reanalysis-era5-single-levels-monthly-means?tab=overview); Sea ice extent: **NSIDC** (https://doi.org/10.7265/N5K072F8); Sea thickness and ice volume: **PIOMAS** ice sea (http://psc.apl.uw.edu/research/projects/arctic-sea-ice-volume-anomaly/data/model grid). PIOMAS-20C reconstruction (https://psc.apl.uw.edu/research/projects/piomas-20c/) and CS2SMOS gridded products Sea surface **OISST** (https://data.meereisportal.de/data/cs2smos awi/v206/n); properties: (https://www.ncei.noaa.gov/products/optimum-interpolation-sst) and OISSS (https://www.esr.org/dataproducts/oisss/overview/); Oceanographic profiles: WOA2023 (https://www.ncei.noaa.gov/products/world-ocean-atlas), EN.4.2.2 objective analyses (https://www.metoffice.gov.uk/hadobs/en4/): Sea surface height: ORAS5 (https://cds.climate.copernicus.eu/datasets/reanalysis-oras5?tab=overview); Ocean heat content: **IAP** Surface albedo: (http://www.ocean.iap.ac.cn/ftp/cheng/IAPv4.2 Ocean heat content 0 6000m/); CLARA-A3 (https://doi.org/10.5676/EUM SAF CM/CLARA AVHRR/V003). In situ observational profiles from four key Arctic regions (the western/eastern Eurasian Basin, the Chukchi Sea, and the Beaufort Sea), thermohaline profiles and Atlantic Water core temperature outputs from five OMIP2 ensemble groups, and Atlantic Water core temperature and depth observational benchmarks, are described in the main text. Detailed metadata specifications and data access instructions for these datasets are provided in the corresponding references cited therein.

Author contributions. XL led the manuscript writing and paper analysis. XL, YC, and HD performed the installation, configuration, and execution of the E3SMv2-MPAS model. HW, KR, and YW were primarily responsible for the conceptualization of the study and manuscript revisions. All co-authors reviewed and commented on the final version of the manuscript.

Competing interests. No competing interests are present.







Acknowledgments. The public availability of different observational data sets and reanalysis data used in this work is a great help for model development, so the efforts of respective working groups are appreciated. This work was supported by the National Key R&D Program of China (Grant No. 2021YFC3101503), the Science and Technology Innovation Program of Hunan Province (Grant No.2022RC3070), the National Natural Science Foundation of China (Grant No. 42305176), the Hunan Provincial Natural Science Foundation of China (Grant No. 42276205) and the Hunan Provincial Science and Technology Innovation Leading Talent Fund.

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
