# Peer review of "Evaluating the E3SMv2-MPAS ocean-sea ice coupled unstructured model in the Arctic: Atlantification processes and systematic biases"

_EGUsphere, 2025_

## Author Comment (AC3)

**Authors point-to-point responds Community Comment #2 to egusphere-2025-2665**

Please find the author's responses in black below the reviewer's comments in blue. The italicized text within quotation marks indicates the proposed revisions in the revised manuscript.

1. Why do the authors particularly reply on the two time periods of 1960-1980 and 1995-2000? Line 160: I dont understand how it can be verified through overlapping period consistency checks (1995–2020).

(1) Regarding the choice of time periods (1960–1980 and 1995–2020):

We appreciate your question regarding the selection of time periods. The division into these two specific periods is based on the following scientific rationale:

- 1995–2020 serves as the core validation period in this study. This interval represents the satellite era, which provides a wealth of remote sensing sea ice data, diverse reanalysis products, and in situ observations, allowing for robust validation of the model's performance. Furthermore, this period covers a phase of accelerated global warming, making it critical for evaluating climate model capabilities.
- 1960–1980 is used as a representative historical period prior to the satellite era, enabling an examination of the model's performance under earlier climatic conditions. Additionally, comparing this period with 1995–2020 helps illustrate the model's response to differing climatic forcings, facilitating a comparative analysis.
- Simulating the entire period from 1960 to 2020 would require prohibitive computational resources, which are beyond the current capacity of our project. Therefore, conducting segmented simulations represents an optimal strategy for balancing scientific objectives with feasibility.

(2) Regarding the question you raised near L160

Regarding the mention of "overlapping period consistency checks," we apologize for any lack of clarity in the original phrasing. We aimed to convey that, under computational constraints, we prioritized the satellite era (1995–2020) for high-confidence validation due to the abundance of observational data (satellite, reanalysis, and in situ) available during this time. The term "consistency check" specifically refers to cross-validation among these multi-source observations. We have revised the manuscript to state this more precisely:

"*Given the prohibitive computational cost of a continuous high-resolution simulation from 1958 to 2020, we adopted a strategic two-period integration scheme to prioritize computational resources for our core analysis period (1995–2020). The model's climatological fidelity during this satellite era is verified using multi-source observational data, ensuring a reliable assessment of both sea ice and ocean variability.*" (P6, L165-168)

2. Line 187: any SSS data below sea ice? Is there any justification?

Thank you for raising this important technical limitation. We confirm that OISSS and other satellite-derived SSS products (e.g., SMAP) cannot provide valid salinity observations under sea ice cover. Their retrieval algorithms actively mask areas with SIC greater than 15%, which are flagged as missing values (NaN) in the datasets.

In the revised manuscript, we continue to use OISSS for validation but explicitly state that SSS comparisons are performed only for open-water areas (SIC < 15%). A spatial mask is applied using the native missing value flags (NaN) from OISSS. At each monthly evaluation time step, statistical calculations are performed only on grid points where valid data exists in both the model output and the observations; regions with sea ice cover (OISSS missing values) are excluded from all analyses.

The vague description in the original Section 2.2.2 has been replaced with a precise statement (P8, L233):

"*For sea surface salinity (SSS), ...*" has been changed to:

"*For open-water sea surface salinity (SSS) validation (SIC<15%), ...*"

3. The major issue is the explanation of the causal analysis throughout the manuscript.  For example, the major conclusion of "These systematic biases may be attributed to three principal sources: inadequate representation of eddy dynamics, limitations in mixing parameterizations, and insufficient resolution of cross-scale interactions in key gateways (e.g., Fram Strait) " is not convincing.  Any sensitivity experiments can be considered to support the findings?

We sincerely thank you for this critical comment. Regarding the statement in the original abstract ("These systematic biases may be attributed to three principal sources: inadequate representation of eddy dynamics, limitations in mixing parameterizations, and insufficient resolution of cross-scale interactions in key gateways (e.g., Fram Strait)."), we provide the following detailed clarifications and context, which have been incorporated into the relevant sections of the revised manuscript. We also acknowledge the limitation of not being able to conduct definitive sensitivity experiments.

(1) "inadequate representation of eddy dynamics"

1) E3SMv2-MPAS failed to capture the observed significant decadal freshening signal in the upper ocean of the Amerasian Basin during the 2000s–2010s (Fig. 13 in the revised manuscript). We have added supporting evidence and references at the relevant location (P23, L502-510):

"*The simulated salinity biases may be related to the use of an inappropriately high and constant isopycnal diffusion coefficient (κ=300 m²/s) in the GM parameterization. This high diffusion coefficient likely results in excessively strong along-isopycnal mixing, which oversmooths horizontal salinity gradient fronts formed by freshwater accumulation (e.g., from melting ice and increased runoff). During the 1970s, when background freshwater signals were relatively weak, the effect of strong diffusion was less pronounced. However, under the strongly increased freshwater input in the 2000s–2010 (Polyakov et al., 2013; Wang et al., 2019), the persistently high κ value continuously and excessively diffused the simulated upper-layer low-salinity anomalies, hindering their realistic accumulation and maintenance in the basin upper layer. As a result, the model significantly underestimates the magnitude of decadal freshening observed in the region.*"

2) The model did not reproduce the observed seasonal variation of the Atlantic Water, characterized by a warmer and thicker in winter compared to summer (Fig. 12). Corroborating discussion has been added in the text (P21, L468-474):

"*This discrepancy may be attributed to the GM parameterization scheme, which models mesoscale eddy effects on heat and salt redistribution through bolus advection and Redi diffusion. In general, the Arctic winter features greater mixed layer depth and weaker stratification due to brine rejection during sea ice formation and wind-driven stirring (Peralta-Ferriz and Woodgate, 2015). These processes promote eddy penetration, increasing the efficiency of vertical heat transport. In contrast, strengthened stratification in summer restricts the vertical scale of eddies and reduces heat transfer. However, the GM scheme employs a fixed diffusion coefficient, which prevents it from capturing the seasonal variability modulated by stratification changes.*"

(2) "limitations in mixing parameterizations"

1) Co-located biases in SST, SSS, and SIC in the Barents and Greenland Seas. Supporting discussion has been added (P17-18, L404-413):

"*In the Greenland and Barents Seas, systematic underestimation of SST and SSS (Figs. 8c and 9c) coincides with overestimation of SIC (Figs. 3 and 4). These regions are situated within the marginal ice zone, where strong surface wind stress facilitates the transfer of energy to deeper ocean layers through the excitation of near-inertial oscillations and associated turbulent mixing processes*

*(D'Asaro, 1985). This discrepancy may be attributed to the model's potential overestimation of this downward energy transfer. Similarly, Zhu et al. (2022) reported that in the equatorial Pacific cold tongue region, the KPP scheme overestimates downward turbulent heat flux, leading to a cold bias in both upper-ocean and sea surface temperatures. A primary reason for these biases lies in the scheme's reliance on a single Richardson number (Ri) relationship for parameterization. Although this approach captures instability conditions in stratified shear flows, it is insufficient to uniquely determine turbulent states and mixing intensities (Zhu et al., 2022), thus limiting its performance in complex dynamic environments.*"

2) Misplaced warming layer in the Eurasian Basin: Observational data indicate warming occurred primarily in the upper~500m on decadal scales, whereas the model simulated the warming predominantly in the 200–1000m layer. Relevant evidence is now discussed in the manuscript (P23, L489-497):

"*These discrepancies may be partly attributed to biases in the representation of vertical processes. As indicated by sensitivity experiments such as those of Liang & Losch (2018), enhanced vertical mixing could promote upward heat transport from AW, potentially causing cooling at intermediate depths (200–900 m). Our model uses a relatively low background diffusivity ($1.0 \times 10^{-5}$ m²/s), which remains constant across time periods despite evidence that Arctic amplification and Atlantification in the 2000s–2010s (Polyakov et al., 2017, 2025; Rantanen et al., 2022; Richards et al., 2022; Shu et al., 2022) may have strengthened vertical mixing compared to the 1970s. The model's failure to represent this temporal increase in mixing efficiency might have limited upward heat transfer, confining warming mainly to intermediate and deeper layers—consistent with the underestimation of shallow warming and exaggerated deep response seen in our simulations.*"

(3) "insufficient resolution of cross-scale interactions in key gateways (e.g., Fram Strait)"

We acknowledge that the original phrasing might have been ambiguous. Our intended meaning was that shortcomings in both vertical mixing and mesoscale eddy parameterizations could potentially be mitigated by increased resolution, particularly in narrow yet critical gateway regions like Fram Strait.

To address this more thoroughly, we have added a new section in the Discussion (Section 5.2: Sources of Systematic Biases and Trade-offs Between Resolution and Parameterizations; P37, L775-798):

"*Analyses in Section 3 not only discussed the simulation biases of E3SMv2-MPAS but also traced their potential origins. For most biases, the primary causes can be attributed*

*to inadequacies in physical parameterizations. First, the inadequate representation of eddy dynamics is a key source. For instance, the underestimation of freshening in the Amerasian Basin may result from the use of a fixed eddy diffusivity ($\kappa=300$ m²/s in the Arctic), which oversmooths salinity fronts. Similarly, the model's failure to capture the seasonal variability of the Atlantic Water layer likely stems from the invariant $\kappa$ in the GM scheme, which cannot respond to the seasonal cycle of sea ice retreat and associated changes in stratification. Second, limitations in vertical mixing parameterizations act as another key source. The coordinated biases in SST, SSS, and SIC in the Greenland and Barents Seas, for example, may arise from the inherent limitations of the KPP scheme's single Ri-based approach in defining turbulent states and mixing intensities within complex dynamic environments. Additionally, the misrepresentation of the warming layer in the Eurasian Basin could be linked to inappropriate background diffusion coefficients within the KPP framework.*

*Increasing model resolution presents an effective pathway to reduce reliance on empirical parameterizations by more directly resolving key physical processes, such as mesoscale eddies. Enhanced resolution can, to some extent, mitigate the inaccuracies of existing schemes. For instance, studies have shown that higher resolution improves the simulation of the Atlantic Water layer's temperature, thickness, spatial distribution, and its decadal warming trends (Wang et al., 2024). However, the small Rossby radius of deformation (often ⩽3 km) in the Arctic (Veneziani et al., 2022) implies that even with computationally feasible resolution increases, critical processes (e.g. mesoscale eddies, vertical mixing, and ice-ocean interactions) may remain under-resolved (Chassignet et al., 2020; Wang et al., 2018). Therefore, the development of more advanced physical parameterizations remains imperative. It is noteworthy that resolution increases have proven effective in improving the simulation of volume, heat, and freshwater transports through critical gateways such as the Fram Strait and Davis Strait (Wang et al., 2024). The Fram Strait, in particular, serves as a pivotal channel for Atlantic heat influx into the Arctic Ocean (Herbaut et al., 2022; Pnyushkov et al., 2021). In conclusion, we propose that a cost-effective strategy involves targetedly increasing resolution in key gateway regions while concurrently refining parameterizations for mesoscale eddies and vertical mixing.*"

Due to significant constraints in computational resources, we have been unable to perform the sensitivity experiments that would be ideal for conclusively proving the sources of the biases. Therefore, the attributions discussed above are currently supported by evidence from the model-observation comparison and references to existing literature. Acknowledging that these attributions remain suggestive rather than proven, we have removed the specific claim regarding the three sources from the abstract. The supporting evidence and reasoning related to these potential sources have been showed in the discussion of biases within Section 3, and in the new Section 5.2 of the Discussion, where

we present them as plausible explanations and areas for future investigation, not as definitive conclusions.

---

## Author Comment (AC4)

**Authors point-to-point responds Community Comment #1 to egusphere-2025-2665**

Please find the author's responses in black below the reviewer's comments in blue. The italicized text within quotation marks indicates the proposed revisions in the revised manuscript.

The main comment is about the state of the E3SMv2-MPAS simulations described and evaluated in this manuscript. The authors mentioned (lines 152-156) that they used the file 'ocean.ARRM60to10.180715.nc' as the MPAS initial condition for E3SMv2-MPAS. That initial condition came from a very short adjustment run (5 days only) with standalone MPAS-ocean that in turn started from rest and Polar science center Hydrographic Climatology (PHC) climatological temperature and salinity, and therefore does not represent a spun-up ocean state. For E3SM-Arctic-OSI, we ran 3 consecutive JRA-55 cycles to achieve a more adjusted state for the ocean and sea ice models, and we analyzed the climatological results over the third cycle only, specifically over the last 12 or 30 years of the simulation (years 148-177 or 166-177; see Figs. 3, 4, 5a, 6, 10-15, 16c,d in Veneziani et al. 2022). Our understanding is that the E3SMv2-MPAS was run for, and evaluated over only one JRA-55 cycle. If that is the case, we wonder 1) whether the analyzed fields are adjusted or not, and also possibly too close to the PHC climatology for a fair evaluation of model performance, and 2) whether it is fair to compare the E3SMv2-MPAS results with more adjusted model states (as done in Figs. 8-10 and 16).

Thank you very much for your valuable feedback and for pointing out the issue regarding the initial state in the E3SMv2-MPAS simulation. In response to your comments and those of the other reviewers, we have revised the manuscript to correct the description of the initial conditions and have provided more detailed simulation configuration information. Please refer to (P6, L169-173): "*The MPAS-Ocean component was initialized from a pre-processed state (ocean.ARRM60to10.180715.nc). This state was derived from a prior short-term (5-day) adjustment run of the standalone ocean model, which itself started from a state of rest with three-dimensional temperature and salinity fields prescribed from the PHC. Consequently, this initial condition provides a dynamically adjusted and physically consistent starting point for our coupled simulation, mitigating the initial shock that would otherwise occur from a purely cold start.*"

We fully agree that completing three cycles of JRA55 forcing and using the results from the last few decades for evaluation would be an ideal approach to obtain a sufficiently equilibrated model state. However, due to constraints in computational resources and

funding, we are currently unable to perform such long-term simulations. In response to the two specific questions you raised, we provide the following clarifications:

1. Regarding your second question, i.e., "whether it is fair to compare the E3SMv2-MPAS results with more adjusted model states (as done in Figs. 8–10 and 16)": We agree that such a comparison may indeed be problematic. Accordingly, we have removed the comparative analyses involving Figures 8, 10, and Table 1 from the original manuscript.

2. Regarding your first question, i.e., "whether the analyzed fields are adjusted or not, and also possibly too close to the PHC climatology for a fair evaluation of model performance": Multiple previous studies have indicated that when using the PHC initial field along with CORE-II or JRA55 forcing in ocean simulations, surface and upper-ocean variables can reach a quasi-equilibrium state within a relatively short time period. For example:

    o Wekerle et al. (2013), using FESOM with CORE-II forcing for a 1958–2007 simulation starting from PHC conditions, analyzed surface variables and freshwater content in the 0–500 m layer after a 10-year spin-up, focusing on the subsequent 40 years.

    o Wang et al. (2018) found that the temperature and salinity structures within the 0–1000 m layer largely reached equilibrium within 20–30 years in a simulation spanning 1950–2009.

    o Wang, Shu, Bozec, et al. (2024) noted that due to computational constraints, many high-resolution OMIP2 models completed only one JRA55 cycle (1958–2018), with their evaluation periods often centered on 1971–2000—i.e., approximately 14 years after initialization.

    Our simulation covers the period 1958–2020, with the main evaluation focused on 1995–2020. The initial conditions for this period were taken from the December 1981 output, meaning the model had already been integrated for 24 years. Although the deep ocean is far from equilibrium, the surface variables (sea ice, sea surface temperature, and salinity) and the upper Atlantic Water layer (above 1000 m), which are the focus of this study, had largely stabilized during this interval, supporting the robustness of our analysis.

    We have added relevant clarifications in the manuscript. Please see (P7, L178-190):

    "*To begin the simulation for our main analysis period (1995–2020), we used the model state from December 1981 as the initial conditions for January 1995. This 13-year gap (1982–1994) was a strategic choice to conserve computational*

*resources while ensuring physical consistency in the key variables of interest. This computational strategy is motivated by the fact that, under forcings such as CORE-II or JRA55 and when initialized with PHC hydrography, upper-ocean and surface variables are known to reach quasi-equilibrium within a few decades, as demonstrated in several previous studies. For instance, Wang et al. (2018) reported that temperature and salinity in the upper 1000 m reached near-equilibrium within 20–30 years. Wekerle et al. (2013) began their analysis of surface variables and freshwater content in the 0–500 m layer after a 10-year initialization in a 1958–2007 simulation using FESOM under CORE-II forcing. Likewise, in the analysis of multiple high-resolution OMIP2 models simulating the full 1958–2020 period under JRA55 forcing, Wang, Shu, Bozec, et al. (2024) focused their evaluation on the period 1971–2000—commencing approximately 14 years after the model initialization. In our simulation, the 24-year spin-up from 1958 to 1981 is largely sufficient for the adjustment of surface fields (e.g., sea ice, surface temperature, and salinity) and Atlantic Water layer (above 1000 m), which are the focus of this study. Although the deep ocean remains far from equilibrium, the targeted variables had largely stabilized by 1981.*"

3. Regarding the comparison with OMIP2 model results (original Figs. 9, 16, and Table 2), we note that these models also completed only one JRA55 cycle, with evaluations typically starting around 14 years after initialization (e.g., 1971–2000). To ensure a fair comparison, we have moved the relevant comparative content to the Discussion section (Section 5.1: Comparison with OMIP2 Models under Diverse Grid Configurations and Resolutions) and restricted the comparison period to 1995–2020. For details, please refer to (P34-35, L732-773).

4. Furthermore, in the Discussion section (Section 5.3: Limitations of the Experimental Design), we have elaborated on the limitations of not completing three full JRA55 cycles (P38, L808-813): "*Furthermore, since only a single JRA55 forcing cycle was applied, the deep ocean and some physical quantities may not have fully departed from the influence of the initial PHC hydrographic fields or reached complete equilibrium. This could potentially affect the stability and initial-condition independence of the simulation results. In subsequent work, given sufficient resources, we plan to carry out at least three full JRA55 forcing cycles to promote more complete adjustment of the ocean state, reduce dependence on initial conditions, and thereby enable a more comprehensive and robust evaluation of the climate performance of the E3SMv2-MPAS.*"

We deeply appreciate your insightful comments, which have significantly improved the quality of our manuscript.

References:

Wang, Q., Wekerle, C., Danilov, S., Wang, X., and Jung, T.: A 4.5 km resolution Arctic Ocean simulation with the global multi-resolution model FESOM 1.4, Geosci. Model Dev., 11, 1229–1255, https://doi.org/10.5194/gmd-11-1229-2018, 2018.

Wang, Q., Shu, Q., Bozec, A., Chassignet, E. P., Fogli, P. G., Fox-Kemper, B., Hogg, A. McC., Iovino, D., Kiss, A. E., Koldunov, N., Le Sommer, J., Li, Y., Lin, P., Liu, H., Polyakov, I., Scholz, P., Sidorenko, D., Wang, S., and Xu, X.: Impact of increased resolution on Arctic Ocean simulations in Ocean Model Intercomparison Project phase 2 (OMIP-2), Geosci. Model Dev., 17, 347–379, https://doi.org/10.5194/gmd-17-347-2024, 2024.

Wekerle, C., Wang, Q., Danilov, S., Jung, T., and Schröter, J.: The Canadian Arctic Archipelago throughflow in a multiresolution global model: Model assessment and the driving mechanism of interannual variability, J. Geophys. Res.-Oceans, 118, 4525–4541, https://doi.org/10.1002/jgrc.20330, 2013.

More specific comments are included in the following.

1. Line 99: Please add reference to our most recent paper on the E3SM-Arctic fully coupled configuration with E3SMv2.1: Huo et al. 2025 (https://doi.org/10.1029/2024MS004726).

   Thank you for your suggestion. We have cited the relevant literature at the recommended location. Please see P3, L97: "(Huo et al., 2024; Ringler et al., 2013)".

2. Line 126: it seems that E3SMv2-MPAS uses the same MPAS mesh configuration as in Veneziani et al. 2022. Here, it would be good to clarify whether that is the case or whether the mesh is different.

   You are absolutely correct—the grid used in E3SMv2-MPAS is the same as that used in Veneziani et al. (2022). A clarification has been provided at P4, L128-129: *"... a meridional transition from 60 km resolution in the Southern Hemisphere to 10 km in the Arctic domain (hereafter 60to10 km, same as Veneziani et al. (2022); Fig. 1a)."*

3. Line 141: the 'spatially varying GM' was actually implemented prior to the development of the E3SM-Arctic-OSI configuration. Nevertheless, the current manuscript does not set GM=0 in the Arctic as we did in Veneziani et al. 2022, so maybe here one could say: "Similarly to what was done in Veneziani et al. 2022, we adopt a spatially varying…" and then include the sentence that is now on lines 212-215.

   Thank you for your suggestion. We have revised the text accordingly (P5, L143-146): *"For mesoscale eddy representation, similarly to what was done in Veneziani et al. (2022), we implement a spatially varying Gent-McWilliams (GM) parameterization, incorporating both bolus advection and Redi isopycnal diffusion components (Gent and Mcwilliams, 1990)."*

   Additionally, since we have now removed the comparative evaluation between E3SMv2-MPAS and E3SM-Arctic-OSI, the sentence originally numbered L212–215 does not appear in the revised manuscript.

4. Line 146: Huo et al. 2025 mentioned above could also be cited here.

   We appreciate you bringing the recent literature to our attention. We have now cited this work at the suggested location. Please see P6, L156-157: *"... have been comprehensively documented in Turner et al. (2022), Golaz et al. (2022) and Huo et al. (2024)."*

5. Line 159: Please clarify how the simulation was restarted after the 1981–1994 gap. What initial condition was used?

Thank you for highlighting this key point, which was not clearly stated in our original manuscript. We have added a detailed description of the simulation design at the end of Section 2.1 (P7, L178-204).

To begin the simulation for our main analysis period (1995-2020), we used the model state from December 1981 as the initial conditions for January 1995. This 13-year gap (1982–1994) was a strategic choice to conserve computational resources while ensuring physical consistency in the key variables of interest.

The validity of this strategy is supported by the following considerations:

1) Previous modeling experience: Under forcing such as JRA55, upper-ocean and surface variables typically reach quasi-equilibrium within a few decades. Our 24-year spin-up (1958–1981) is consistent with or longer than the adjustment periods used in many published studies, ensuring that the upper ocean and sea ice—the focus of this work—are adequately equilibrated.

2) Physical rationale: The upper ocean and sea ice adjust much more rapidly than the deep ocean and are primarily governed by contemporary atmospheric forcing rather than initial conditions. Thus, any initial imbalance due to the 13-year gap is quickly overcome by the realistic forcing applied from 1995 onward.

3) Model validation: The output from the 1995–2020 simulation shows physically consistent behavior, with key variables such as sea surface temperature and sea ice concentration aligning closely with observations and reanalysis products shortly after initialization, showing no persistent bias.

6. Line 205 (and more generally for the manuscript): the comparison with CMIP6 experiments seems unfair to us because those simulations are fully coupled. In addition, the comparison with the OMIP models and E3SM-Arctic-OSI should be done over similar (more adjusted) oceanic states. Finally, for the Veneziani et al. 2022 paper, we also provided data from a E3SMv1-LR-OSI simulation, which could be used for comparison to the E3SMv2-MPAS simulations here.

We are grateful for this important comment. We fully agree that a direct comparison between E3SMv2-MPAS and either fully coupled CMIP6 models or a well-spun-up E3SM-Arctic-OSI would be scientifically unfair. As mentioned in our response to the main comment, we have therefore removed all comparative analyses involving CMIP6 and E3SM-Arctic-OSI (including the original Fig. 10).

We recognize that a rigorous comparison should be conducted over equivalent time periods after completing simulations of the same length (e.g., three JRA55 cycles). Should computational resources allow in the future, we plan to carry out the full simulation and perform a comprehensive and equitable comparison of E3SMv2-MPAS results with E3SM-Arctic-OSI and the E3SMv1-LR-OSI you mentioned.

7. Line 230 (and Figs. 3-4): We assume these are annual sea ice quantities. Wouldn't it be better to show seasonal sea ice climatologies?

You are entirely correct—the original Figures 3 and 4 depicted annual sea ice quantities. Following your suggestion, we have now separated the evaluations of sea ice concentration and thickness into winter (December–February) and summer (June–August). Please refer to the revised Figures 3–5 and the related analysis (P10-12, L284-307).

8. As mentioned above, the comparison with E3SM-Arctic-OSI in section 3.3 (Fig. 10) should be over the same time period. For the Veneziani et al. 2022 paper, we provided climatologies for both E3SM-Arctic-OSI and E3SM-LR-OSI over years 148-177 and 166-177 (end of third JRA-55 cycle). A similar time frame should be used for E3SMv2-MPAS.

Thank you for reiterating this point. We strongly concur with your view that any inter-model comparison should be conducted over the same time period to be scientifically valid.
Given the current computational constraints, which prevent us from completing three full JRA55 cycles, we agree that such a comparison in the original Section 3.3 was not appropriate. We have therefore decided to remove the original Figure 10 and all associated comparative analysis.
We have also explicitly acknowledged this limitation in the Discussion section of the paper (P38, L808-813).

9. Similarly to other reanalysis products, JRA55 is known to overestimate surface air temperatures over Arctic sea ice (e.g., Batrak and Müller 2019, https://doi.org/10.1038/s41467-019-11975-3; see their Fig. 3d). These warm biases can propagate into the ocean component by modifying surface fluxes—particularly enhancing downward longwave radiation and reducing sensible heat loss—potentially leading to underestimated sea ice growth and overestimated upper-ocean temperatures. A brief discussion on how such forcing biases might influence ocean stratification, mixed layer depth, or Atlantification could be included in the Conclusions section.

We greatly appreciate your suggestion. A new subsection (Section 5.4: Limitations from the Atmospheric Forcing: The JRA55 Warm Bias) has been added, specifically addressing the limitations arising from the warm bias in JRA55. This addition significantly enhances the academic rigor of the paper. For details, please see (P38, L815-827):

"*The JRA55 reanalysis forcing data employed in E3SMv2-MPAS exhibits a known warm bias over the central Arctic deep basin (Batrak & Müller, 2019). This bias may systematically suppress sea ice growth and induce upper-ocean warming in the simulation by enhancing downward longwave radiation and reducing oceanic sensible heat loss. Therefore, the overestimated SST (Fig. 8c) and underestimated summer SIC (Fig. 4e–g) simulated in the central Arctic basin may be partially attributable to the inherent bias in the forcing data, rather than solely to inaccuracies in the model's physical processes.*

*The enhanced ice melt driven by this warm bias releases additional freshwater, leading to a stronger and shallower freshwater layer (a more pronounced halocline) in the surface ocean, which significantly strengthens the stratification stability of the upper ocean. This inhibits vertical mixing between layers and impedes the upward heat transfer from the warmer, saltier AW below. This bias may partly explain the overestimation of the intermediate Atlantic Water layer temperature alongside the underestimation of the mixed-layer temperature (Fig. 12e). Future work will consider employing alternative reanalysis products or applying bias-correction methods to better constrain the impact of forcing uncertainties on simulation results.*"

---

## Author Response (AR1)

**Referee #1:**

2.5

This study evaluates the performance of E3SMv2-MPAS on the Arctic simulation, with emphasize on sea ice concentration, sea ice thickness, SST/SSS, AW layer temperature and depth. The authors find that the E3SMv2-MPAS is superior on some aspects and also some limitations are identified. In general, the manuscript is well written, easy to follow, and the scientific significance of the manuscript is guaranteed. I hope the following comments are useful for the authors to revise their manuscript, most of them are minor, major comments are marked with \*\*\*.

Thank you very much for your thorough review and positive feedback on our manuscript. We are grateful for your insightful comments and suggestions, which have greatly helped us improve the quality of our paper. We have carefully addressed each of your points in the revised manuscript. Our point-by-point responses are detailed below.

Please find the author's responses in black below the reviewer's comments in blue. The italicized text within quotation marks indicates the proposed revisions in the revised manuscript. The page and line numbers mentioned in the responses below refer to the clean version of the revised manuscript. Please note that the line numbers here differ slightly from the previous version submitted for community review, as the manuscript has been adjusted to meet GMD formatting standards.

- 1. L11: change "Arctic sea ocean-sea ice system" to "Arctic ocean-sea ice system" Thanks, the phrase has been changed on P1, L10.
  - 2. L14: please clarify what "multi-scale" refers to.

Thank you for requesting clarification. The term "multi-scale" in our context primarily refers to temporal scales, encompassing a broad range from seasonal to interannual-decadal variability. We have revised the abstract (P1, L13-14): "... a simulation framework capable of resolving processes from seasonal to decadal timescales."

3. L23-24: "These systematic biases may be attributed to three principal sources ..... in key gateways". This speculation should be supported by some evidences in the main text.

Regarding the statement in the original abstract ("These systematic biases may be attributed to three principal sources: inadequate representation of eddy dynamics, limitations in mixing parameterizations, and insufficient resolution of cross-scale interactions in key gateways (e.g., Fram Strait)."), we provide the following detailed clarifications and context, which have been incorporated into the relevant sections of the revised manuscript. However, it is important to note that, due to computational constraints, the attribution of these biases primarily relied on insights from the existing literature. Acknowledging that these attributions remain suggestive rather than proven, we have removed the specific claim regarding the three sources from the abstract.

(1) "inadequate representation of eddy dynamics"

- 1) E3SMv2-MPAS failed to capture the observed significant decadal freshening signal in the upper ocean of the Amerasian Basin during the 2000s–2010s (Fig. 13 in the revised manuscript). We have added supporting evidence and references at the relevant location (P23, L494-502):
  - "The simulated salinity biases may be related to the use of an inappropriately high and constant isopycnal diffusion coefficient ( $\kappa$ =300 m² s⁻¹) in the GM parameterization. This high diffusion coefficient likely results in excessively strong along-isopycnal mixing, which oversmooths horizontal salinity gradient fronts formed by freshwater accumulation (e.g., from melting ice and increased runoff). During the 1970s, when background freshwater signals were relatively weak, the effect of strong diffusion was less pronounced. However, under the strongly increased freshwater input in the 2000s–2010 (Polyakov et al., 2013; Wang et al., 2019), the persistently high  $\kappa$  value continuously and excessively diffused the simulated upper-layer low-salinity anomalies, hindering their realistic accumulation and maintenance in the basin upper layer. As a result, the model significantly underestimates the magnitude of decadal freshening observed in the region."
- 2) The model did not reproduce the observed seasonal variation of the Atlantic Water, characterized by a warmer and thicker in winter compared to summer (Fig. 12). Corroborating discussion has been added in the text (P21, L460-466):
  - "This discrepancy may be attributed to the GM parameterization scheme, which models mesoscale eddy effects on heat and salt redistribution through bolus advection and Redi diffusion. In general, the Arctic winter features greater mixed layer depth and weaker stratification due to brine rejection during sea ice formation and wind-driven stirring (Peralta-Ferriz and Woodgate, 2015). These processes promote eddy penetration, increasing the efficiency of vertical heat transport. In contrast, strengthened stratification in summer restricts the vertical scale of eddies and reduces heat transfer. However, the GM scheme employs a fixed diffusion coefficient, which prevents it from capturing the seasonal variability modulated by stratification changes."
- (2) "limitations in mixing parameterizations"
- 1) Co-located biases in SST, SSS, and SIC in the Barents and Greenland Seas. Supporting discussion has been added (P17-18, L396-405):

"In the Greenland and Barents Seas, systematic underestimation of SST and SSS (Figs. 8c and 9c) coincides with overestimation of SIC (Figs. 3 and 4). These regions are situated within the marginal ice zone, where strong surface wind stress facilitates the transfer of energy to deeper ocean layers through the excitation of near-inertial oscillations and associated turbulent mixing processes (D'Asaro, 1985). This discrepancy may be attributed to the model's potential overestimation of this downward energy transfer. Similarly, Zhu et al. (2022) reported that in the equatorial Pacific cold tongue region, the KPP scheme overestimates downward turbulent heat flux, leading to a cold bias in both upper-ocean and sea surface temperatures. A primary reason for these biases lies in the scheme's reliance on a single Richardson number (Ri) relationship for parameterization. Although this approach captures instability conditions in stratified shear flows, it is insufficient to uniquely determine turbulent states and mixing intensities (Zhu et al., 2022), thus limiting its performance in complex dynamic environments."

- 2) Misplaced warming layer in the Eurasian Basin: Observational data indicate warming occurred primarily in the upper~500m on decadal scales, whereas the model simulated the warming predominantly in the 200–1000m layer. Relevant evidence is now discussed in the manuscript (P22-23, L481-489):
  - "These discrepancies may be partly attributed to biases in the representation of vertical processes. As indicated by sensitivity experiments such as those of Liang & Losch (2018), enhanced vertical mixing could promote upward heat transport from AW, potentially causing cooling at intermediate depths (200–900 m). Our model uses a relatively low background diffusivity (1.0×10-5 m² s-1), which remains constant across time periods despite evidence that Arctic amplification and Atlantification in the 2000s–2010s (Polyakov et al., 2017, 2025; Rantanen et al., 2022; Richards et al., 2022; Shu et al., 2022) may have strengthened vertical mixing compared to the 1970s. The model's failure to represent this temporal increase in mixing efficiency might have limited upward heat transfer, confining warming mainly to intermediate and deeper layers consistent with the underestimation of shallow warming and exaggerated deep response seen in our simulations."
- (3) "insufficient resolution of cross-scale interactions in key gateways (e.g., Fram Strait)"
- We acknowledge that the original phrasing might have been ambiguous. Our intended meaning was that shortcomings in both vertical mixing and mesoscale eddy parameterizations could potentially be mitigated by increased resolution, particularly in narrow yet critical gateway regions like Fram Strait.

To address this more thoroughly, we have added a new section in the Discussion (Section 5.2: Sources of systematic biases and trade-offs between resolution and parameterizations; P37, L766-790):

"Analyses in Section 3 not only discussed the simulation biases of E3SMv2-MPAS but also traced their potential origins. For most biases, the primary causes can be attributed to inadequacies in physical parameterizations. First, the inadequate representation of eddy dynamics is a key source. For instance, the underestimation of freshening in the Amerasian Basin may result from the use of a fixed eddy diffusivity ( $\kappa$ =300 m² s¹ in the Arctic), which oversmooths salinity fronts. Similarly, the model's failure to capture the seasonal variability of the AW layer likely stems from the invariant  $\kappa$  in the GM scheme, which cannot respond to the seasonal cycle of sea ice retreat and associated changes in stratification. Second, limitations in vertical mixing parameterizations act as another key source. The coordinated biases in SST, SSS, and SIC in the Greenland and Barents Seas, for example, may arise from the inherent limitations of the KPP scheme's single Ri-based approach in defining turbulent states and mixing intensities within complex dynamic environments. Additionally, the misrepresentation of the warming layer in the Eurasian Basin could be linked to inappropriate background diffusion coefficients within the KPP framework.

Increasing model resolution presents an effective pathway to reduce reliance on empirical parameterizations by more directly resolving key physical processes, such as mesoscale eddies. Enhanced resolution can, to some extent, mitigate the inaccuracies of existing schemes. For instance, studies have shown that higher resolution improves the simulation of the AW layer's temperature, thickness, spatial distribution, and its decadal warming trends (Wang et al., 2024). However, the small Rossby radius of deformation (often  $\leq 3$ km) in the Arctic (Veneziani et al., 2022) implies that even with computationally feasible resolution increases, critical processes (e.g. mesoscale eddies, vertical mixing, and iceocean interactions) may remain under-resolved (Chassignet et al., 2020; Wang et al., 2018). Therefore, the development of more advanced physical parameterizations remains imperative. It is noteworthy that resolution increases have proven effective in improving the simulation of volume, heat, and freshwater transports through critical gateways such as the Fram Strait and Davis Strait (Wang et al., 2024). The Fram Strait, in particular, serves as a pivotal channel for Atlantic heat influx into the Arctic Ocean (Herbaut et al., 2022; Pnyushkov et al., 2021). In conclusion, we propose that a cost-effective strategy involves targetedly increasing resolution in key gateway regions while concurrently refining parameterizations for mesoscale eddies and vertical mixing."

4. L27: change "components" to "area"

We thank you for this suggestion. The change has been made accordingly (P1, L25).

5. L37-38: "Liang and Losch, 2018; Tian et al., 2022" are based on regional ice-ocean model, not climate model.

Thank you for correctly pointing this out. The cited references "Duarte et al., 2020; Hinrichs et al., 2021; Liang and Losch, 2018a; Tian et al., 2022; Wassmann et al., 2015" were indeed based on regional models. We have updated them to more appropriate citations: "Dörr et al., 2021; Hinrichs et al., 2021; Rieke et al., 2023; Shu et al., 2022". (P2, L35)

6. L48: change "and" to "to"

Corrected as suggested (P2, L45).

7. L53: change "seafloor regions" to "seafloor"

Corrected as suggested (P2, L50).

8. L60: change "shifts" to "shift"

Corrected as suggested (P2, L57).

9. L79: change "temporal" to "spatial"?

We sincerely thank you for catching this error. You are correct; "temporal" should be "spatial". This has been corrected (P3, L76).

10. L95: change "FESOM's" to "FESOM"

This has been corrected in the revised manuscript (P3, L92).

11. L96: delete "then"

This word has been deleted (P3, L93).

12. L130-132: "The North Atlantic sector ...... in the Gulf Stream extension region". Please rephrase this sentence.

The sentence has been rephrased for clarity: "... the North Atlantic sector is strategically refined, transitioning from 20 km to 10 km resolution earlier than the Pacific to guarantee at least 15 km resolution in the Gulf Stream extension region (~40° N; Veneziani et al. (2022) ..." (P5, L130-131)

13. L134: "subpolar North Pacific sector adjacent to the Arctic Ocean"?

Thank you for identifying this careless error in our description. The sentence has been corrected to: "... the North Pacific sector maintains computational efficiency while achieving approximately 10 km resolution in its subpolar region adjacent to the Arctic Ocean (north of 50° N)." (P5, L132-133)

**14. L139: how long is the sea ice dynamic step? The same to ocean dynamic step?**

We apologize for the imprecise original description. The sea ice dynamic time step is 15 minutes. The manuscript has been revised to state this clearly (P5, L138-139): "For sea ice, we employed a 15-minute dynamic time step and a 30-minute thermodynamic time step (a 2:1 ratio)."

**15. L157: rapidly**

We thank you for this correction. In response to the feedback received from other reviewers, we have determined that the original phrasing containing this term was imprecise. Consequently, the sentence has been removed from the revised manuscript.

16. \*\*\*L159-161: "The simulation periods ...... period consistency checks (1995–2020)". If I understand correctly, you derive the simulation of 1995-2020 using JRA55 forcing during 1995-2020 but initialized at the latest model state of 1980. It is seldom to see such design of model simulation. As the intermediate years only span 15 years, I suggest the authors conduct a continuously simulation from 1960 to 2020.

We greatly appreciate your deep insight into our simulation design. Your understanding is perfectly accurate: to initiate the 1995–2020 JRA55-forced simulation, we used the model state from the end of 1981 (generated by a prior 1958–1981 JRA55-forced simulation) as the initial conditions, without performing any model integration for the period 1982–1994.

We fully share your concern regarding continuity. However, a continuous and complete simulation from 1958 to 2020 was computationally prohibitive under our resource constraints. Given these limitations, we prioritized ensuring a simulation for our core analysis period (1995–2020), which benefits from the richest observational data.

From a physical mechanism perspective, the potential impact of this initialization approach for the 1995–2020 simulation is likely confined to the very beginning of our analysis period. The core focus of our study – the surface and upper ocean, along with sea ice – exhibits much shorter adjustment timescales compared to the deep ocean and is predominantly governed by the contemporaneous atmospheric forcing. Thus, disequilibrium introduced by the initial conditions (the state from the end of 1981) would be rapidly overwritten and adjusted by the realistic, synchronous atmospheric forcing applied from 1995 onward. This approach is physically justified and analogous to the common practice in ocean modeling of initializing with climatological mean states (e.g., PHC), which similarly relies on atmospheric forcing to constrain the model's interannual variability. (This reasoning is also presented in the manuscript: P7, L191-195.)

The model output initialized from the 1981 state also demonstrates physically consistent behavior during the 1995–2020 period, further supporting the validity of this approach.

The temporal evolution of key diagnostic variables – including sea surface temperature (Fig. 8d) and sea ice-related variables (Fig. 7) – shows that the simulation quickly aligns with the observed/reanalysis trajectory after 1995, with no persistent systematic bias. Spatial distributions of these variables are also in good agreement with evaluation datasets (Figs. 3–5, 8a–c), and the long-term trends from 1995 to 2020 closely match those in the references (Fig. 7). These results, which will be discussed in detail in the following sections, indicate that the initialization from 1981 did not adversely affect the simulation of central climate features during the study period. (This reasoning is also presented in the manuscript: P7, L198-204.)

Furthermore, we have explicitly acknowledged the limitation of this simulation design in the Discussion section (Section 5.3: Limitations of the experimental design), and have committed to performing a full continuous simulation from 1958 to 2020 in future work when computational resources allow:

"Due to computational resource constraints, this study adopted a two-phase simulation strategy with non-consecutive time periods: first, the model was integrated from 1958 to 1981, and the final state of this period was used as the initial condition to directly start the simulation for the 1995–2020 period. Although this approach effectively reduced computational costs, and both previous studies and our model diagnostics indicate that key upper-ocean and sea ice variables had reached a quasi-equilibrium state by 1981, skipping the continuous integration of the 1982–1994 period may introduce certain limitations. For instance, the simulation of some medium- to long-term fluctuations or memory-dependent processes might be affected. Should computational resources allow in the future, we will perform a continuous simulation from 1958 to 2020 to more accurately reproduce the evolution of the climate system." (P37-38, L792-799)

17. \*\*\*Section 2.1: there is no detailed information of sea ice model provided here. Please specify sea ice thermodynamics and dynamics in this configuration. As section 3.1 relates to sea ice validation, sea ice model description is necessary.

We thank you for this helpful comment. We have now added a detailed description of the sea ice thermodynamics and dynamics configurations used in MPAS-Seaice in Section 2.1 (P5-6, L147-154). An excerpt is provided below:

"MPAS-Seaice builds upon the core numerical and physical framework of the Los Alamos Sea Ice Model (CICE). The dynamics are governed by the elastic-viscous-plastic (EVP) rheology, with the internal ice stress divergence operator adapted for MPAS's unstructured polygonal mesh (Turner et al., 2022). Sea ice and tracer transport are handled by an incremental remapping scheme (Lipscomb and Ringler, 2005), adapted for polygonal cells. The thermodynamics and vertical column physics remain consistent with CICE (Turner et al., 2022). The configuration includes the "mushy layer" thermodynamics for vertical heat transfer, the delta-Eddington shortwave radiation scheme, a level-ice melt pond parameterization, ice thickness distribution mechanics, and transport in thickness space (Petersen et al., 2019)."

**18. L197: EN.4.2.2 dataset**

Thank you for spotting this typo. This error has been corrected (P9, L241).

- 19. L244: from Figure 3a, "systematic winter overestimation" is caused partly by positive sea ice bias in the southern Greenland Sea and south extension of sea ice cover in the Barents Sea, suggesting upper ocean temperature bias in these regions. "moderate summer underestimation" may be related to inaccurate ice-albedo feedback and melt pond dynamics.
- We appreciate this insightful comment. Our original text only described the biases observed in the time series without discussing their potential sources.
  - As you rightly pointed, the "systematic winter overestimation" is partly caused by the positive sea ice bias in the southern Greenland Sea and the southward extension of sea ice cover in the Barents Sea (Fig. 3 in the revised manuscript). Following your suggestion regarding potential upper ocean temperature bias, we have added an analysis of winter SST spatial biases (Fig. S3), which indeed reveals significant cold biases in precisely these regions.
  - According to the interannual time series from 1995 to 2020, E3SMv2-MPAS overestimates the summer minima (Fig. 7a). Combined with the spatial maps of summer SIC (Fig. 4), the overestimation is primarily located in the Greenland Sea, Barents Sea, East Siberian-Laptev Seas, and Beaufort Sea. We further analyzed summer albedo (Fig. S4) and found that the model simulates higher albedo in these specific regions, leading to reduced absorbed shortwave radiation and consequently an overestimation of sea ice.
- We have incorporated this discussion on the potential sources of the seasonal SIC biases
  into the revised manuscript (P15, L343-348): "Consistent with NSIDC, simulated SIC and
  SIE exhibit certain seasonal biases. The systematic winter overestimation, attributable to
  positive SIC biases in the southern Greenland Sea and southward-expanded ice cover in
  the Barents Sea (Fig. 3e), coinciding with pronounced cold SST biases in these regions
  (Fig. S3). During summer, E3SMv2-MPAS overestimates the seasonal minimum (Fig. 7a–
  b), particularly in the Greenland Sea, Barents Sea, East Siberian-Laptev Seas, and
  Beaufort Sea (Fig. 4e). These regions also exhibit elevated surface albedo values (Fig.
  S4), reducing absorbed shortwave radiation and contributing to the sea ice
  overestimation."

20. L249: "overestimated seasonal variability amplitudes" relates to sea ice thermodynamics, as no sea ice thermodynamics information is provided, it is hard to judge its causality.

The phrase "overestimated seasonal variability amplitudes" was originally a description for the 1960–1980 period, which might still be within the model's spin-up phase and lacks robust observational data for detailed attribution. Consequently, we have removed this statement from the revised manuscript. Our analysis of bias sources now focuses primarily on the well-observed 1995–2020 period, as described in response to your previous comment.

21. \*\*\*L258-269, 279-281: The modeled SIT has large biases in the Beaufort Gyre region, suggesting potential upper ocean thermal biases in the Beaufort Gyre. The author could check whether the ocean-ice heat flux over the Beaufort Gyre region is reasonable.

Thank you for this excellent suggestion.

Given the scarcity of direct ocean-ice heat flux measurements in the Arctic, we analyzed the 0-100m ocean heat content (OHC) instead. A comparison with the IAP observational dataset reveals a general underestimation of OHC in the Beaufort Gyre region, which could contribute to the overestimation of SIT there.

We have included this supporting analyzes in the revised manuscript (P12-13, L317-322):

"The model overestimates SSH in the Beaufort Sea, suggesting an erroneously enhanced ice convergence. Additionally, the simulated OHC in the 0–100 m layer is underestimated in this region (Fig. 6d–f), which may further contribute to the positive SIT bias. Thus, the persistent 0.5–1 m positive bias in the Beaufort Sea is hypothesized to originate from an overestimated intensity of the Beaufort Gyre and associated upper-ocean thermal biases in E3SMv2-MPAS, which then may impede the realistic export of sea ice through the north of Canadian Archipelago and east of Greenland."

Additionally, following your suggestion, we examined the ocean-ice heat flux in the Beaufort Gyre region (shown in the Fig. 1 below). A broad comparison with Figure 2 from Zhong et al. (2022) suggests the overall magnitude is reasonable, though it might suffer from a systematic low bias consistent with the OHC analysis.

Figure 1. Gridded average ocean-ice heat flux during (a) 2006–2012 and (b) 2013–2018.

**Reference:**

Zhong, W., Cole, S. T., Zhang, J., Lei, R., and Steele, M.: Increasing Winter Ocean-to-Ice Heat Flux in the Beaufort Gyre Region, Arctic Ocean Over 2006–2018, Geophysical Research Letters, 49, e2021GL096216, https://doi.org/10.1029/2021GL096216, 2022.

22. Figure 4d: given the known biases of the PIOMAS SIT, I suggest the author additionally validate the modeled sea ice volume against that derived from the CS2SMOS SIT from 2012.

We are grateful for this suggestion. We have replaced the comparison with ICESat by validating the modeled SIT against the CS2SMOS SIT product for winter (December-February) over the period 2011-2019. This new analysis confirms that E3SMv2-MPAS systematically overestimates SIT in the Canadian Archipelago and Greenland coastal regions. The corresponding text has been updated (P12, L306-312):

"Considering PIOMAS's known limitations in overestimating thin ice while underestimating thick ice (Laxon et al., 2013; Schweiger et al., 2011), additional validation using CS2SMOS data (Ricker et al., 2017) is conducted (Fig. S1). Consistent with previous findings, PIOMAS exhibits underestimation in regions with thicker sea ice, such as north of the Canadian Archipelago and east of Greenland (Fig. S1e). Similarly, E3SMv2-MPAS shows pronounced positive biases relative to CS2SMOS in areas including the northern Canadian Archipelago, the southern Canadian Basin, and the Beaufort Sea (Fig. S1d), aligning with the bias pattern identified in comparisons with PIOMAS (Fig. 5c), thereby corroborating the spatial reliability of PIOMAS-indicated biases."

23. L347: Please specify the define of AW layer thickness.

Thank you for your comment. However, since the spin-up duration of E3SMv2-MPAS is considerably shorter than that of fully-coupled CMIP6 models, a direct comparison between the two would not be scientifically equitable. Therefore, the analysis section containing this sentence has been removed from the revised manuscript.

24. L347-349: This sentence needs to be rephrased as "Amerasian Basin" is not in Figure 8.

Thanks for pointing this out. The mentioned sentence has been removed in the new version.

25. L362-363: "E3SMv2-MPAS maintains systematic temperature overestimation (~0.5C average)". This statement is not appropriate for the other three regions.

We agree that this statement was not accurate or rigorous enough. It has been revised to (P19, L431-432): "(~1°C in the western Eurasian Basin, ~0.3°C in the Chukchi Sea and the Beaufort Sea) ..."

26. L364-365: systematic salinity underestimation only occurs in western and eastern Eurasian Basin.

Thank you. This sentence has been removed along with the deleted CMIP6 comparison section.

27. L390: ~ 0.5 C at 800 m?

We apologize for this careless error. You are correct; it has been changed to "the same depth" for accuracy (P35, L740).

28. L417: I understand from section 2 that both the E3SMv1 and E3SMv2 use the same KPP. Why the vertical mixing scheme in E3SMv2 is refined?

We thank you for your careful reading. Similar to CMIP6, E3SMv2-MPAS was run with one cycle of JRA55 forcing, whereas E3SMv1 was run with three full cycles. This difference in experimental setup makes a direct comparison between the two models inappropriate. Accordingly, the comparison with E3SMv1, including the discussion of the KPP scheme, has been removed from the revised manuscript. We apologize for any inaccuracy in our original wording.

29. Figure 11: please clarify this figure is conducted over the whole Arctic Basin or Eurasian Basin or Amerasian Basin?

We apologize for not specifying the region. The figure represents a pan-Arctic Basin average. This clarification has been added to the figure caption: "Figure 11. For the Arctic Basin, ...". (P20, L440)

30. \*\*\*L466-467, 469-470: "likely modulated by differential ocean-ice feedbacks and cross-basin transport dynamics", "indicating limitations in AW transport pathways and heat redistribution", such speculations are too arbitrary, it's better to avoid using such speculations.

Thanks for this feedback. We agree that these speculations were too arbitrary without stronger direct evidence. Consequently, the phrases "likely modulated by differential ocean-ice feedbacks and cross-basin transport dynamics" and "indicating limitations in AW transport pathways and heat redistribution" have been removed from the revised manuscript.

31. L488-490: see the previous comment.

As noted above, the related text has been deleted.

32. L519: delete "the" before "Fram Strait". "Similar negative deviations (-0.5 C)"

The sentence containing "the Fram Strait" pertained to the attribution of model biases and was not sufficiently rigorous; it has therefore been removed from the revised manuscript. The phrasing "Similar negative deviations (-0.5 °C)" has been corrected (P30, L641).

33. L527-530: This conclusion can not derived from the observations STRICTLY.

You are absolutely right. We have erred on the side of caution and removed this sentence.

34. L582: a detailed description of the thermal linkage framework between the upper and intermediate ocean layers is needed here.

We sincerely thank you for this insightful comment. We agree that a more precise description of the methodology is necessary. The core of this framework is the analysis of spatiotemporal correlation (both instantaneous and lagged) between temperature at 5 m and 400 m depths.

We have revised the manuscript accordingly to provide a clearer and more detailed description. The specific changes can be found in P32, L690-692: "A diagnostic framework based on spatiotemporal correlation analysis is established to quantify the thermal linkage between the upper (10 m) and intermediate (AW core layer, 400 m) ocean layers."

**Referee #2:**

The manuscript titled 'Evaluating the E3SMv2-MPAS Ocean-Sea Ice Coupled Unstructured Model in the Arctic: Atlantification Processes and Systematic Biases' presents a coupled ocean—sea ice model based on the E3SMv2-MPAS framework from the Energy Exascale Earth System Model, designed for Arctic sea ice and ocean simulations. The model features a high resolution of 10 km in the Arctic Ocean. According to the model validation, it demonstrates good performance in simulating Arctic sea ice and ocean conditions. Utilizing these simulations, the authors further identify a fundamental regime shift in intermediate-to-surface thermal coupling mechanisms in the Arctic under climate warming. In my assessment, the manuscript requires major revision. I offer the following suggestions to enhance its quality:

We sincerely appreciate the thorough and constructive review of our manuscript. We are grateful for the positive recognition, as well as for the insightful comments aimed at further improving the study. We have carefully considered all the suggestions provided and have revised the manuscript accordingly to address each point raised. Below, we provide a point-by-point response to the comments.

Please find the author's responses in black below the reviewer's comments in blue. The italicized text within quotation marks indicates the proposed revisions in the revised manuscript. The page and line numbers mentioned in the responses below refer to the clean version of the revised manuscript. Please note that the line numbers here differ slightly from the previous version submitted for community review, as the manuscript has been adjusted to meet GMD formatting standards.

1. I concur with CC1's comments. It is not entirely appropriate to compare the E3SMv2-MPAS results with those from E3SM-Arctic-OSI, CMIP6, and OMIP models, due to significant differences in integration lengths and associated model drifts. These discrepancies undermine the fairness of direct comparison. I recommend that the authors remove the model intercomparison from the main text and instead address relevant points briefly in the discussion section.

Thank you very much for your comments. We have revised the manuscript accordingly, shifting the focus of the analyze to the evaluation and comparison between E3SMv2-MPAS and observations/reanalysis data. And we have removed the comparisons with CMIP6 and E3SMv1 (formerly in Figs. 8, 10, and Table 1).

Furthermore, as noted by Wang et al. (2024), due to the substantial computational resources required for high-resolution simulations, high-resolution studies within the OMIP2 framework have typically considered only one JRA55 cycle (1958–2018).

Therefore, in the Discussion section, we have briefly compared and evaluated E3SMv2-MPAS against these high-resolution OMIP2 models that also completed only one JRA55 cycle. For details, please refer to Section 5.1, titled "Comparison with OMIP2 models under diverse grid configurations and resolutions" (P34-36, L724-765).

**Reference:**

- Wang, Q., Shu, Q., Bozec, A., Chassignet, E. P., Fogli, P. G., Fox-Kemper, B., Hogg, A. McC., Iovino, D.,
  Kiss, A. E., Koldunov, N., Le Sommer, J., Li, Y., Lin, P., Liu, H., Polyakov, I., Scholz, P., Sidorenko, D.,
  Wang, S., and Xu, X.: Impact of increased resolution on Arctic Ocean simulations in Ocean Model
  Intercomparison Project phase 2 (OMIP-2), Geosci. Model Dev., 17, 347–379, https://doi.org/10.5194/gmd-17-347-2024, 2024.
- 2. The model configuration requires further clarification. As the simulations do not adhere to the standard OMIP protocol, more detailed information regarding the model integration setup should be provided.
  - We sincerely appreciate your suggestion. We have primarily supplemented the detailed design of the simulation period in Section 2.1 (P6-7, L164-207):
- "Given the prohibitive computational cost of a continuous high-resolution simulation from 1958 to 2020, we adopted a strategic two-period integration scheme to prioritize computational resources for our core analysis period (1995–2020). The model's climatological fidelity during this satellite era is verified using multi-source observational data, ensuring a reliable assessment of both sea ice and ocean variability.
- The MPAS-Ocean component was initialized from a pre-processed state
   (ocean.ARRM60to10.180715.nc). This state was derived from a prior short-term (5-day)
   adjustment run of the standalone ocean model, which itself started from a state of rest
   with three-dimensional temperature and salinity fields prescribed from the PHC.
   Consequently, this initial condition provided a dynamically adjusted and physically
   consistent starting point for our coupled simulation, mitigating the initial shock that
   would otherwise occur from a purely cold start. In contrast, the MPAS-Seaice component
   was initialized from an idealized, uniform ice cover. A 1-meter thick ice layer with 100%
   concentration was prescribed on all ocean grid points between 60° S and 70° N, with
   zero initial snow depth and stationary ice velocity. This simple state allowed the sea ice
- zero initial snow depth and stationary ice velocity. This simple state allowed the sea ice
  cover to evolve self-consistently in response to the model's atmospheric forcing and ocean
  coupling from the beginning of the simulation. Following this spin-up phase, the full
  interannual JRA55 forcing was applied from 1958 to 1981.
  - To begin the simulation for our main analysis period (1995–2020), we used the model state from December 1981 as the initial conditions for January 1995. This 13-year gap (1982–1994) was a strategic choice to conserve computational resources while ensuring 475 physical consistency in the key variables of interest. This computational strategy was motivated by the fact that, under forcings such as CORE-II or JRA55 and when initialized with PHC hydrography, upper-ocean and surface variables are known to reach quasiequilibrium within a few decades, as demonstrated in several previous studies. For instance, Wang et al. (2018) reported that temperature and salinity in the upper 1000 m 480 reached near-equilibrium within 20–30 years. Wekerle et al. (2013) began their analysis of surface variables and freshwater content in the 0-500 m layer after a 10-year initialization in a 1958–2007 simulation using FESOM under CORE-II forcing. Likewise, in the analysis of multiple high-resolution the Ocean Model Intercomparison Project Phase 2 (OMIP2) models simulating the full 1958–2020 period under JRA55 forcing, 485 Wang, Shu, Bozec, et al. (2024) focused their evaluation on the period 1971–2000 – commencing approximately 13 years after the model initialization. In our simulation, the 24-year spin-up from 1958 to 1981 is largely sufficient for the adjustment of surface fields (e.g., sea ice, surface temperature, and salinity) and AW layer (above 1000 m). which are the focus of this study. Although the deep ocean remains far from equilibrium, 490 the targeted variables had largely stabilized by 1981.

From a physical perspective, the potential impact of this initialization approach for the 1995–2020 simulation is expected to be short-lived. The upper ocean and sea ice (the primary focus of this study), adjust much more rapidly than the deep ocean, and their evolution is predominantly governed by contemporaneous atmospheric forcing rather than by the initial conditions. Therefore, the disequilibrium introduced by the initial condition from 1981 would be rapidly overwritten and adjusted by the realistic, synchronous atmospheric forcing applied from 1995 onward.

Therefore, initializing the 1995 run from the 1981 output allows a computationally efficient hot start and ensures that the model is in an appropriate state for evaluating the 1995–2020 period.

The model output initialized from the 1981 state also demonstrates physically consistent behavior during the 1995–2020 period, further supporting the validity of this approach. The temporal evolution of key diagnostic variables – including sea surface temperature (Fig. 8d) and sea ice-related variables (Fig. 7) – shows that the simulation quickly aligns with the observed/reanalysis trajectory after 1995, with no persistent systematic bias. Spatial distributions of these variables are also in good agreement with evaluation datasets (Figs. 3–5, 8a–c), and the long-term trends from 1995 to 2020 closely match those in the references (Fig. 7). These results, which will be discussed in detail in the following sections, indicate that the initialization from 1981 did not adversely affect the simulation of central climate features during the study period.

Accordingly, our primary evaluation focuses on the performance of E3SMv2-MPAS during the period 1995–2020. In addition, a comparative assessment of the 1960–1980 period is also included to briefly examine the decadal variability of key ocean and sea ice variables and to verify the model's capability under distinctly different climatic backgrounds."

3. Regarding sea ice validation, while sea ice concentration and thickness are evaluated, I encourage the authors to also include assessments of sea ice extent, volume, and their long-term trends.

We are grateful for your recommendation. In the revised manuscript, we have added analysis on the evaluation of sea ice extent and volume. Please see Figure 7 and the related content on P13-15, L327-358.

- 4. Given that this is a model evaluation study, I suggest a more comprehensive evaluation of the Arctic Ocean simulations. Key metrics should include Arctic Ocean freshwater content and its trend, as well as volume, heat, and freshwater fluxes through major Arctic gateways. These aspects are critical for assessing the model's performance in simulating Arctic Ocean climate.
- Following your advice, we have added two new sections of the main text: Section 3.4 "Freshwater content spatiotemporal variability" (P24-26, L521-557) and Section 3.5 "Gateway transports: volume, heat, and freshwater" (P26-29, L559-631). These sections present a comparative evaluation between E3SMv2-MPAS results and observational data.
- The added analyses demonstrate that E3SMv2-MPAS faithfully reproduces both the spatial distribution and long-term trend of Arctic freshwater content (Fig. 15). Furthermore, it accurately simulates volume, heat, and freshwater transports through key Arctic gateways, capturing their observed magnitudes and essential variability trends (Fig. 16).

---

## Author Response (AR2)

Thank you for the significantly improved revised manuscript. The reviewers have suggested the following minor adjustments, which I agree with:

Thanks for your positive feedback on our revised manuscript. We are very grateful to you and reviewers for the careful reading and constructive comments, which have helped further enhance the quality of our paper.

We have addressed all the points raised, and our responses are provided below.

5

25

Please reduce the colorbar range in Figure 3e, 3f, 3g, 4e, 4f, 4g to amplify the anomalies on the maps.

We sincerely appreciate this suggestion. The colorbar ranges in the specified figures have been adjusted accordingly as suggested.

Line 50: change "extending to seafloor" to "extending to the seafloor".

Line 54: change "with no model replicating post-2000 acceleration in AW warming" to "with no model replicating the post-2000 acceleration in AW warming".

Line 185: change "high-resolution the Ocean Model Intercomparison Project Phase 2 (OMIP2) models" to "high-resolution Ocean Model Intercomparison Project Phase 2 (OMIP2) models".

Line 229: change "dataset, represents a long-term climate data record" to "dataset, which represents a long-term climate data record".

Line 421: change "E3SMv2-MPAS can successfully reproduces observed vertical temperature structure" to "E3SMv2-MPAS can successfully reproduce observed vertical temperature structure".

Line 602: change "A study by Schauer et al. (2004) show annual mean net volume transport" to "A study by Schauer et al. (2004) shows annual mean net volume transport".

Thank you and reviewer #2 for the thorough and attentive review. All the suggested grammatical and phrasing corrections have been carefully incorporated into the revised manuscript.

Specifically, we have updated the text at the indicated lines, with line numbers referring to the track-changes file:

"extending to **the** seafloor" (Line 50);

"the post-2000 acceleration" (Line 54);

```
"high-resolution Ocean Model Intercomparison Project..." (Line 185);

"dataset, which represents..." (Line 229);

"can successfully reproduce" (Line 423);

"Schauer et al. (2004) shows" (Line 604).
```

**After those changes, we can then proceed to publish. Thanks again!**

After these changes, we sincerely hope the manuscript now meets the journal's standards for publication. Once again, we extend our deepest gratitude to you and reviewers for your insightful comments and unwavering support throughout the review process. We look forward to your final confirmation.